

# Geographical and altitudinal distribution of *Brachycephalus* (Anura: Brachycephalidae) endemic to the Brazilian Atlantic Rainforest

Marcos R. Bornschein[1,2], Carina R. Firkowski[3], Ricardo Belmonte-Lopes[2,3,4], Leandro Corrêa[2], Luiz F. Ribeiro[2,3,5], Sérgio A.A. Morato[6], Reuber L. Antoniazzi-Jr.[7], Bianca L. Reinert[2,8], Andreas L.S. Meyer[3,4], Felipe A. Cini[3] and Marcio R. Pie[2,3,4]

[1] Instituto de Biociências, Universidade Estadual Paulista, São Vicente, São Paulo, Brazil
[2] Mater Natura—Instituto de Estudos Ambientais, Curitiba, Paraná, Brazil
[3] Departamento de Zoologia, Universidade Federal do Paraná, Curitiba, Paraná, Brazil
[4] Programa de Pós-Graduação em Zoologia, Departamento de Zoologia, Universidade Federal do Paraná, Curitiba, Paraná, Brazil
[5] Escola de Saúde, Pontifícia Universidade Católica do Paraná, Curitiba, Paraná, Brazil
[6] STCP Engenharia de Projetos Ltda., Curitiba, Paraná, Brazil
[7] Red de Ecoetología, Instituto de Ecología A.C., Xalapa, Veracruz, Mexico
[8] Laboratório de Biodiversidade, Conservação e Ecologia de Animais Silvestres, Departamento de Zoologia, Universidade Federal do Paraná, Curitiba, Paraná, Brazil

Corresponding author
Marcos R. Bornschein,
bornschein.marcao@gmail.com

## ABSTRACT

Mountains of the Brazilian Atlantic Forest can act as islands of cold and wet climate, leading to the isolation and speciation of species with low dispersal capacity, such as the toadlet species of the genus *Brachycephalus*. This genus is composed primarily by diurnal species, with miniaturized body sizes (<2.5 cm), inhabiting microhabitats in the leaf litter of montane forests. Still, little is known about the geographical distribution, altitudinal range, and ecological limits of most *Brachycephalus* species. In this study, we review the available data on the geographical and altitudinal distribution of *Brachycephalus* based on occurrence records compiled from literature and museums, both for the genus as a whole and separately for the three recently proposed groups of species (*ephippium*, *didactylus*, and *pernix*). The final ensemble dataset comprised 333 records, 120 localities, 28 described species, and six undescribed ones. Species were recorded in six relief units, the richest of which being the Serra do Mar, with 30 species. When the Serra do Mar is subdivided into three subunits, Northern, Central and Southern Serra do Mar, the number of species increase from north to the south, with records of six, nine, and 16 species, respectively. We were able to estimate the extent of occurrence of nearly half of the described species, and the resulting estimates indicate that many of them show remarkably small ranges, some of which less than 50 ha. *Brachycephalus* species are present from sea level to roughly 1,900 m a.s.l., with the highest richness being found between 751 and 1,000 m a.s.l. (21 spp.). The species with the broadest altitudinal range were *B. didactylus* (1,075 m) and *Brachycephalus* sp. 1 (1,035 m), both in the *didactylus* group, and *B. ephippium* (1,050 m), of the *ephippium* group. The broadest altitudinal amplitude for species of the *pernix* group was recorded for *B. brunneus* (535 m). The lowest altitudinal records for the *pernix* group were at

845 m a.s.l. in the state of Paraná and at 455 m a.s.l. in the state of Santa Catarina. The altitudinal occurrence in the *pernix* species group seems to decrease southward. Syntopy between species is also reviewed.

## INTRODUCTION

*Brachycephalus* Fitzinger is a genus of miniaturized diurnal toadlets (usually <2.5 cm in snout-vent length) that inhabit the forest floor of montane regions along the Atlantic Rainforest of southeastern and southern Brazil (*Izecksohn, 1971*; *Giaretta & Sawaya, 1998*; *Pombal Jr, Wistuba & Bornschein, 1998*; *Napoli et al., 2011*; *Pie et al., 2013*; see *Rocha et al. (2000)* and *Pombal Jr & Izecksohn (2011)* for reports of nocturnal activity). *Brachycephalus* presents direct development and a reduction in the number and size of digits (*Hanken & Wake, 1993*; *Pombal Jr, 1999*; *Yeh, 2002*). Some species are aposematic (yellow, orange or yellow with light red), of which some were confirmed as harboring neurotoxins (tetrodotoxin and analogues; *Sebben et al., 1986*; *Pires Jr et al., 2002*; *Pires Jr et al., 2003*; *Schwartz et al., 2007*).

Recently, *Kaplan (2002)* proposed that the genus *Psyllophryne* Izecksohn should be considered as a junior synonym of *Brachycephalus*. *Psyllophryne* included two species, *P. didactyla* and *P. hermogenesi*, "with dorsal surfaces brown, tiny body sizes (snout-vent length 8.6–10.2 mm), body leptodactyliform, and long-distance jumpers" (*Napoli et al., 2011*). There are currently 29 described species of *Brachycephalus* (*Frost, 2015*; see *Pie & Ribeiro, 2015*; *Ribeiro et al., 2015*), and although 20 of them were described within the last 10 years, it is likely that many more species are still waiting to be described. Moreover, ecological data on species of the genus *Brachycephalus* is still limited. To date it only includes data on the breeding behavior of *B. ephippium* (*Pombal Jr, Sazima & Haddad, 1994*; *Pombal Jr, 1999*); vocal activity of *B. hermogenesi* (*Verdade et al., 2008*); advertisement calls of *B. ephippium* (*Pombal Jr, Sazima & Haddad, 1994*), *B. hermogenesi* (*Verdade et al., 2008*), *B. pitanga* (*Araújo et al., 2012*), *B. tridactylus* (*Garey et al., 2012*), and *B. crispus* (*Condez et al., 2014*); abundance estimates of *B. didactylus* (*Van Sluys et al., 2007*; *Almeida-Santos et al., 2011*; *Rocha et al., 2013*; *Siqueira et al., 2014*) and its diet (*Almeida-Santos et al., 2011*); the diet of *B. brunneus* (*Fontoura, Ribeiro & Pie, 2011*), *B. garbeanus* (*Dorigo et al., 2012*), and *B. pitanga* (*Oliveira & Haddad, 2015*); and microhabitat preference of *B. garbeanus* (*Dorigo et al., 2012*).

A recently-proposed phylogenetic hypothesis for the genus indicated their separation into three main clades, with still some uncertainty regarding relationships within each clade (*Clemente-Carvalho et al., 2011b*). Based on the species studied by *Clemente-Carvalho et al. (2011b)*, one clade included *B. didactylus* and *B. hermogenesi*; a second clade, included *B. brunneus*, *B. ferruginus*, *B. izecksohni*, *B. pernix*, and *B. pombali*; and a third clade included *B. alipioi*, *B. ephippium*, *B. garbeanus*, *B. nodoterga*, *B. pitanga*, *B. toby*, and *B. vertebralis*. Two morphological traits—the presence/absence of dermal

co-ossification and body shape—were used to distinguish three groups of species (*Ribeiro et al., 2015*), namely *ephippium*, *didactylus*, and *pernix* groups. The first was defined by the presence of dermal co-ossification and bufoniform body shape and includes the following species, distributed in southeastern Brazil (from the state of Espírito Santo south to the state of São Paulo): *B. alipioi*, *B. bufonoides*, *B. crispus*, *B. ephippium*, *B. garbeanus*, *B. guarani*, *B. margaritatus*, *B. nodoterga*, *B. pitanga*, *B. toby*, and *B. vertebralis* (*Ribeiro et al., 2015*). The *didactylus* group was defined by the absence of dermal co-ossification and a leptodactyliform body shape and includes *B. didactylus*, *B. hermogenesi*, and *B. pulex* (*Ribeiro et al., 2015*). Finally, the *pernix* group was defined by the absence of dermal co-ossification and bufoniform body shape, and includes the following species, distributed in southern Brazil (states of Paraná and Santa Catarina): *B. auroguttatus*, *B. boticario*, *B. brunneus*, *B. ferruginus*, *B. fuscolineatus*, *B. izecksohni*, *B. leopardus*, *B. mariaeterezae*, *B. olivaceus*, *B. pernix*, *B. pombali*, *B. tridactylus*, and *B. verrucosus* (*Ribeiro et al., 2015*). Later, an additional species from the pernix group was described from southern Brazil (*Pie & Ribeiro, 2015*): *B. quiririensis*. On the other hand, the holotype of *Brachycephalus atelopoides* is currently missing (*Pombal Jr, 2010*) and no population of this species in nature is known. Therefore, given that diagnostic characters could not be inspected, *B. atelopoides* was not assigned to any of the species groups (*Ribeiro et al., 2015*). A recent study by *Padial, Grant & Frost (2014)* indicated that *B. hermogenesi* would be closely related to a species of the *ephippium* group, implying that the *didactylus* and *ephippium* clades were not monophyletic, but the authors used the same dataset as the original *Clemente-Carvalho et al. (2011b)* study. One possibility for this incongruence is that *Padial, Grant & Frost (2014)* only analyzed the concatenated dataset, which could have masked variations among different among loci that were accounted for in species tree analyses by Clemente-Carvalho et al. (*Ribeiro et al., 2015*).

An intriguing aspect of *Brachycephalus* is the high level of microendemism found in most of its species. For instance, extensive field surveys often find species confined to one or a few adjacent mountaintops along the Atlantic Forest (*Pie et al., 2013*), suggesting that their diversification is strongly linked to the topographical characteristics of their habitats. Therefore, to understand the origin and distribution of *Brachycephalus*, it is crucial to integrate species occurrence data with the geographical attributes of the biomes where they are found. The Atlantic Forest, the second largest forested biome of South America, can be divided at least into eight broad relief units, of which three correspond to higher altitude regions, namely plateaus (''Planaltos''), mountain ranges (''Serras''), and escarpments (''Escarpas'') (*IBGE, 1993*). The highest mountains of the Atlantic Forest biome occur in the Serra da Mantiqueira and Serra do Mar. Their formation is associated with tectonic events dating back to the Paleocene, which caused an uplift of an eastern sector of southeastern Brazil and led to a depression in the eastern border. This region was subsequently submerged by the ocean and led to the formation of the Santos basin (*Almeida & Carneiro, 1998*). A second downgrade occurred further inland, establishing the Serra da Mantiqueira (it extends throughout the states of Espírito Santo, Minas Gerais, Rio de Janeiro, and São Paulo; *Almeida & Carneiro, 1998*), which is currently divided by the ''Paraíba do Sul'' river valley into northern and southern rims

(*IBGE, 1993*; *IBGE, 2006*; *MMA, 2016*). The Serra do Mar, formed between 300 and 400 million years before present, contitutes the new eastern border of the Brazilian plateau that was progressively eroded westward, leaving remnants that formed coastal islands (*Almeida & Carneiro, 1998*). This edge of the plateau, called Serra do Mar, was interrupted from the southern state of São Paulo by headward erosion of the Ribeira river valley and from its lower limit in northeastern state of Santa Catarina by headward erosion of the Itajaí Açu river valley (*Almeida & Carneiro, 1998*).

Millions of years of erosion processes have shaped the escarpments and mountains in eastern and southern Brazil. Instead of forming a continuous massif range, several blocks of high elevation isolated by lower areas have been formed, constituting a mountain range. These blocks receive one or more regional names of "Serras," the highest of which being Serra do Caparaó (2,892 m a.s.l.) and Serra do Itatiaia (2,791 m a.s.l.), both part of Serra da Mantiqueira. The Serra do Mar in its northern limit (northern state of Rio de Janeiro) is characterized by several "Serras" with the highest peak reaching up to 2,230 m a.s.l. Its middle section presents a distinctive structure, featuring a high plateau at the border between the states of Rio de Janeiro and São Paulo, which decreases in altitude southward until southeast of São Paulo city. In this region around the São Paulo city, the Serra do Mar is characterized as a border of the plateau (*Almeida & Carneiro, 1998*), reaching 800–900 m a.s.l., whereas in the border between the states of São Paulo and Rio de Janeiro its altitude reaches up to 2,050 m a.s.l. In the state of Paraná, the Serra do Mar is characterized again by several "Serras" with altitudes from 500 to 1,000 m above the plateau (*Maack, 1981*); the highest peak reaches up to 1,800 m a.s.l. (*sensu* Google Earth (satellite image from 2015), or 1,922 m a.s.l., *sensu Maack (1981)*). The Serra do Mar in the state of Santa Catarina has a very short extension (about 45 km in a north/south straight line). The highest altitudes are found only near the border with the state of Paraná, where it reaches up to 1,524 m. Further south, the Serra do Mar becomes increasingly lower, reaching only ∼600 m a.s.l. on the southern edge of occurrence of this relief unit.

The mountainous regions of the Atlantic Forest might have contributed to the formation and distribution limits of *Brachycephalus* in several ways. For instance, mountains could have supported population remnants of many species during warmer periods throughout the Quaternary (e.g., *Carnaval et al., 2009*; *Brunes et al., 2010*; *Thomé et al., 2010*; *Amaro et al., 2012*; see also *Wollenberg et al., 2011*; *Páez-Moscoso & Guayasamin, 2012*, and *Giarla, Voss & Jansa, 2014* for similar scenarios in other montane regions) mainly due to their distinctly colder and wetter climatic conditions. Hence, they played an important role in the diversification of animals with low dispersion capacities, including certain birds (*Mata et al., 2009*) and amphibians (*Cruz & Feio, 2007*; *Brunes et al., 2010*; *Thomé et al., 2010*). This has been confirmed in an investigation of the environmental niches of different *Brachycephalus* species groups, given that they were shown to occupy distinct regions of climatic space (*Pie et al., 2013*). That analysis included most of the described species (as well some that were undescribed) and was based occurrence records associated with species descriptions, museum specimens, and a few comprehensive references. They have also presented new geographical distribution limits for the genus and records of altitude for

all species. However, no study to date has explored explicitly the altitudinal variation and provided estimates of geographical distributions of *Brachycephalus* species.

Understanding the ecological limits, geographical distributions, and altitudinal ranges of *Brachycephalus* species is of considerable importance, particularly to direct more effective conservation actions. In the present study, we address the aforementioned aspects by reviewing the geographical and altitudinal distribution of *Brachycephalus* based on records compiled from literature and museum specimens. In particular, expanding a previous effort by *Pie et al. (2013)*, we now include the entire literature on *Brachycephalus*, which nearly tripled the number of sources in relation to that study. We analyze the geographical distribution and altitudinal amplitude of occurrence of the genus as a whole, as well as separately based on their species groups.

## MATERIALS AND METHODS

### Species records

In an earlier paper (*Pie et al., 2013*), occurrence records of *Brachycephalus* species were obtained from the SpeciesLink portal (www.splink.org.br), as well as from DZUP (Coleção de Herpetologia, Departamento de Zoologia, Universidade Federal do Paraná, state of Paraná, Brazil) and MHNCI (Museu de História Natural Capão da Imbuia - Prefeitura Municipal de Curitiba) collections, and from the species descriptions themselves. For this paper we expanded that dataset by compiling additional records from DZUP and from a broader review of the literature. A few unpublished records obtained by the authors in southern Brazil were also included to improve the dataset. Collection, handling and preservation permits were issued by ICMBIO (22470–1 and 22470–2). We carried out the same careful analysis as in *Pie et al. (2013)* to check the precise geographical coordinates of point records using topographic maps (at different scales), resources of the Google Earth software, including the titles of photographs associated with different locations, and descriptive information of localities in other publications. The final dataset used for the analyses included only those records which coordinate precision index value was between 1.1 and 3.0, according to McLaren et al., reprinted in *Knyazhnitskiy et al. (2000)*. To help further revisions, we also listed occurrence records considered imprecise after our revision (see 'Results'). Although there are methods for georeferencing imprecise locations (e.g., *Wieczorek, Guo & Hijmans, 2004*), we preferred to discard the records that did not meet these standards.

Altitudinal records were either compiled from the literature or, when that information was unavailable, by plotting occurrence records on Google Earth. The altitudinal range of some species were obtained directly from the literature or from the field by the authors. The altitudinal range represents the lower and higher altitudinal occurrence record of a species in a given locality. Although some records had precise locations, they were located on slopes where the exact altitude of the record was difficult to ascertain and therefore were omitted from further analysis, but not from the compiled table, to assist future revisions.

All records were associated with Brazilian relief units (*MMA, 2016*) by simply plotting each record against a map of the limits of different relief units. In two of the major relief

units, we also analyzed our data separately for their subunits as follows: the Serra da Mantiqueira as "Northern Serra da Mantiqueira" (states of Espírito Santo and adjacent Minas Gerais) and "Southern Serra da Mantiqueira" (states of Minas Gerais, Rio de Janeiro, and São Paulo), and the Serra do Mar as "Northern Serra do Mar" (state of Rio de Janeiro), "Central Serra do Mar" (southern state of Rio de Janeiro and state of São Paulo), and "Southern Serra do Mar" (southern state of São Paulo and states of Paraná and Santa Catarina). The Serra da Mantiqueira we subdivided into two sectors because this unit comprises naturally two sectors, isolated by the "Paraíba do Sul" relief unit (*IBGE, 1993*; *IBGE, 2006*; *MMA, 2016*). The Serra do Mar we subdivided into sectors according to the interruption of escarpments (*IBGE, 2006*). Geographical coordinates are based into the WGS84 datum.

## Geographical distribution measurement

We tentatively provide estimates of species geographical distributions as their "extent of occurrence," *sensu IUCN (2012)*, using two methods. The first is an adaptation of "minimum convex polygon" (*Mohr, 1947*), as *Reinert, Bornschein & Firkowski (2007)*, which allowed for changing the shape of the minimum convex polygon, whenever it was possible, to remove areas that are obviously unsuitable for the persistence of the species, such as cities, plantations, bodies of water, and vegetation in the early stages of regeneration, as well areas beyond the altitudinal amplitude record of the species. In the second method, we made polygons by considering the lowest altimetric quota with records of the species, while also omitting unsuitable areas (as above). We made these environmental and altitudinal assessments and produced the resulting polygons using Google Earth Pro 7.1.4.1529. We then measured the areas encompassed by each polygon using GEPath 1.4.5. Although we disregarded unsuitable areas when drawing the polygons, we treated the resulting measurements as "extent of occurrence" and not "area of occupancy" (*sensu IUCN, 2012*) because microhabitat requirements can prevent the species from occurring in some parts of the area within the polygons. Unfortunately, we were unable to determine occurrence polygons of many species when the abovementioned criteria led to obviously unrealistic distribution polygons, primarily for the following reasons: (1) when there were exceedingly few records that were usually very scattered and/or encompassed vast degraded areas, or (2) when the altitudinal range of occurrence of a species based on their altitudinal limits encompassed unreasonably large areas (usually heading west).

## Species groups

All described species in the present compilation were assigned to one of the three species groups by *Ribeiro et al. (2015)*, except for *B. atelopoides* (see above). The present compilation also includes some undescribed species not yet assigned to one of the three groups; all of them were mentioned as new species or potentially new species in the respective source of their records. Two of them were here assigned to the *ephippium* group because they probably represent populations of *B. ephippium* that could be split into two species, one of which from the state of Rio de Janeiro (*Siqueira, Vrcibradic & Rocha, 2013*; see also *Siqueira et al., 2011*) and the other from the state of São Paulo (*Pie et al., 2013*).

Bornschein et al. (2016), *PeerJ*, DOI 10.7717/peerj.2490

**Table 1** Records of *Brachycephalus* spp. Altitude is provided in meters (above sea level).

| Species[a] | Relief units[b] | Group[c] | Locality[d] | State | Altitude | Source [e] |
|---|---|---|---|---|---|---|
| *B. didactylus* | Northern Serra da Mantiqueira | *didactylus* | Monumento Natural Serra das Torres (21°00′04″S, 41°13′17″W), municipality of Atílio Vivácqua | ES | ? (600–900?) | *Oliveira et al. (2012)*, *Oliveira et al. (2013)* |
| *B. didactylus* | Northern Serra do Mar | *didactylus* | Fazenda Santa Bárbara (22°25′17″S, 42°35′01″W), Parque Estadual dos Três Picos, municipality of Cachoeiras de Macacu | RJ | 500–800 | *Siqueira et al. (2009)*, *Almeida-Santos et al. (2011)* |
| *B. didactylus* | Northern Serra do Mar | *didactylus* | Reserva Ecológica de Guapiaçu (22°24′00″S, 42°44′00″W), municipality of Cachoeiras de Macacu | RJ | 300–520 | *Siqueira et al. (2014)* |
| *B. didactylus* | Northern Serra do Mar | *didactylus* | Reserva Ecológica Rio das Pedras (22°59′00″S, 44°06′45″W), municipality of Mangaratiba | RJ | 200–1,110 | This study, *Carvalho-e-Silva, Silva & Carvalho-e-Silva (2008)*, *Almeida-Santos et al. (2011)*, *Rocha et al. (2013)* |
| *B. didactylus* | Northern Serra do Mar | *didactylus* | Sacra Família do Tinguá (22°29′11″S, 43°36′18″W), municipality of Engenheiro Paulo de Frontin | RJ | 600 | *Izecksohn (1971)*, *Pombal Jr (2001)*, *Ribeiro, Alves & Haddad (2005)*, *Alves et al. (2006)*, *Alves et al. (2009)*, *Silva, Campos & Sebben (2007)*, *Verdade et al. (2008)*, *Clemente-Carvalho et al. (2009)*, *Campos (2011)*, *Pombal Jr & Izecksohn (2011)*, *Pie et al. (2013)* |
| *B. didactylus* | Northern Serra do Mar | *didactylus* | Theodoro de Oliveira (first position: 22°22′11″S, 42°33′25″W), Parque Estadual dos Três Picos, municipality of Nova Friburgo | RJ | ? (1,100–1,400?) | This study, *Siqueira et al. (2011)* |
| *B. didactylus* | Northern Serra do Mar | *didactylus* | Tinguá (22°35′51″S, 43°24′54″W), municipality of Nova Iguaçu | RJ | 35 | *Izecksohn (1971)* |
| *B. didactylus* | Northern Serra do Mar | *didactylus* | Vila Dois Rios (23°11′01″S, 44°12′23″W), Ilha Grande, municipality of Angra dos Reis | RJ | 220–240 | This study, *Rocha et al. (2000)*, *Rocha et al. (2001)*, *Van Sluys et al. (2007)* |
| *B. hermogenesi* | Central Serra do Mar | *didactylus* | Corcovado (23°28′08″S, 45°11′29″W), municipality of Ubatuba | SP | ? (70?) | *Giaretta & Sawaya (1998)*, *Verdade et al. (2008)* |
| *B. hermogenesi* | Central Serra do Mar | *didactylus* | Estação Biológica de Boracéia (first position: 23°39′10″S, 45°53′05″W), municipality of Salesópolis | SP | 900 | *Pimenta, Bérnils & Pombal Jr (2007)*, *Verdade et al. (2008)* |
| *B. hermogenesi* | Central Serra do Mar | *didactylus* | Fazenda Capricórnio (23°23′27″S, 45°04′26″W), municipality of Ubatuba | SP | 60 | *Giaretta & Sawaya (1998)*, *Verdade et al. (2008)* |
| *B. hermogenesi* | Central Serra do Mar | *didactylus* | Picinguaba (23°22′21″S, 44°49′53″W), Parque Estadual da Serra do Mar, municipality of Ubatuba | SP | 0–700 | *Giaretta & Sawaya (1998)*, *Pimenta, Bérnils & Pombal Jr (2007)*, *Verdade et al. (2008)*, *Clemente-Carvalho et al. (2009)*, *Pie et al. (2013)* |

Bornschein et al. (2016), *PeerJ*, DOI 10.7717/peerj.2490

**Table 1** (*continued*)

| Species[a] | Relief units[b] | Group[c] | Locality[d] | State | Altitude | Source[e] |
|---|---|---|---|---|---|---|
| *B. hermogenesi* | Central Serra do Mar | *didactylus* | Reserva Biológica do Alto da Serra de Paranapiacaba (23°46′40″S, 46°18′45″W), municipality of Santo André | SP | ? (800?) | *Verdade et al. (2008)* |
| *B. hermogenesi* | Central Serra do Mar | *didactylus* | Reserva Florestal de Morro Grande (23°42′08″S, 46°58′22″W), municipality of Cotia | SP | ? (990?) | *Dixo & Verdade (2006)*, *Verdade et al. (2008)* |
| *B. pulex* | Pré-Litorâneas | *didactylus* | Serra Bonita (15°23′28″S, 39°33′59″W), municipality of Camacan | BA | 800–930 | *Napoli et al. (2011)* |
| *Brachycephalus* sp. 1 | Paranapiacaba | *didactylus* | Caratuval (24°51′17″S, 48°43″43″W), near the Parque Estadual das Lauráceas, municipality of Adrianópolis | PR | 900 | *Firkowski* (*2013*; without species identification), *Pie et al.* (*2013*; as "*Brachycephalus* sp. nov. 1") |
| *Brachycephalus* sp. 1 | Paranapiacaba | *didactylus* | Caratuval (24°51′14″S, 48°42′01″W), Parque Estadual das Lauráceas, municipality of Adrianópolis | PR | 890 | *Pie et al.* (*2013*; as "*Brachycephalus* sp. nov. 1") |
| *Brachycephalus* sp. 1 | Paranapiacaba | *didactylus* | Fazenda Primavera (24°53′08″S, 48°45′51″W), municipality of Tunas do Paraná | PR | 1,060 | *Pie et al.* (*2013*; as "*Brachycephalus* sp. nov. 1") |
| *Brachycephalus* sp. 1 | Paranapiacaba | *didactylus* | Fazenda Thalia (25°30′58″S, 49°40′12″W), municipality of Balsa Nova | PR | 1,025 | *Firkowski* (*2013*; without species identification), *Pie et al.* (*2013*; as "*Brachycephalus* sp. nov. 1") |
| *Brachycephalus* sp. 1 | Paranapiacaba | *didactylus* | Base of the Serra Água Limpa (24°28′52″S, 48°47′12″W), municipality of Apiaí | SP | 920 | This study, DZUP, *Firkowski* (*2013*; without species identification) |
| *Brachycephalus* sp. 1 | Southern Serra do Mar | *didactylus* | Alto Quiriri (26°05′34″S, 48°59′41″W), municipality of Garuva | SC | 240 | *Pie et al.* (*2013*; as "*Brachycephalus* sp. nov. 1") |
| *Brachycephalus* sp. 1 | Southern Serra do Mar | *didactylus* | Castelo dos Bugres (first position: 26°13′47″S, 49°03′20″W), municipality of Joinville | SC | 790–860 | *Pie et al.* (*2013*; as "*Brachycephalus* sp. nov. 1") |
| *Brachycephalus* sp. 1 | Southern Serra do Mar | *didactylus* | Colônia Castelhanos (25°47′58″S, 48°54′40″W), municipality of Guaratuba | PR | 290 | *Cunha, Oliveira & Hartmann* (*2010*; as "*Brachycephalus* aff. *hermogenesi*"), *Oliveira et al.* (*2011*; as "*B. hermogenesi*"), *Pie et al.* (*2013*; as "*Brachycephalus* sp. nov. 1") |
| *Brachycephalus* sp. 1 | Southern Serra do Mar | *didactylus* | Corvo (25°20′17″S, 48°54′56″W), municipality of Quatro Barras | PR | 930 | *Firkowski* (*2013*; without species identification), *Pie et al.* (*2013*; as "*Brachycephalus* sp. nov. 1") |
| *Brachycephalus* sp. 1 | Southern Serra do Mar | *didactylus* | Dona Francisca (26°09′52″S, 48°59′23″W), municipality of Joinville | SC | 150 | *Pie et al.* (*2013*; as "*Brachycephalus* sp. nov. 1") |
| *Brachycephalus* sp. 1 | Southern Serra do Mar | *didactylus* | Estrada do rio do Júlio (26°17′02″S, 49°06′08″W), municipality of Joinville | SC | 650 | *Mariotto* (*2014*; as "*Brachycephalus* sp.") |
| *Brachycephalus* sp. 1 | Southern Serra do Mar | *didactylus* | Fazenda Pico Paraná (25°13′29″S, 48°51′17″W), municipality of Campina Grande do Sul | PR | 1,050 | *Pie et al.* (*2013*; as "*Brachycephalus* sp. nov. 1") |

Bornschein et al. (2016), *PeerJ*, DOI 10.7717/peerj.2490

| Species[a] | Relief units[b] | Group[c] | Locality[d] | State | Altitude | Source [e] |
|---|---|---|---|---|---|---|
| *Brachycephalus* sp. 1 | Southern Serra do Mar | *didactylus* | Mananciais da Serra (25°29′29″S, 48°58′40″W), municipality of Piraquara | PR | 1,050 | This study, DZUP, *Pie et al.* (*2013*; as "*Brachycephalus* sp. nov. 1") |
| *Brachycephalus* sp. 1 | Southern Serra do Mar | *didactylus* | Pico Agudinho (25°36′24″S, 48°43′33″W), Serra da Prata, municipality of Morretes | PR | 385 | *Pie et al.* (*2013*; as "*Brachycephalus* sp. nov. 1") |
| *Brachycephalus* sp. 1 | Southern Serra do Mar | *didactylus* | Recanto das Hortências (25°33′24″S, 48°59′38″W), municipality of São José dos Pinhais | PR | 975 | This study, DZUP |
| *Brachycephalus* sp. 1 | Southern Serra do Mar | *didactylus* | Reserva Particular do Patrimônio Natural Salto Morato (25°10′00″S, 48°17′21″W), municipality of Guaraqueçaba | PR | 45–900 | This study, *Pereira et al.* (*2010*; as "*B. hermogenesi*"), *Santos-Pereira et al.* (*2011*; as "*B. hermogenesi*") |
| *Brachycephalus* sp. 1 | Southern Serra do Mar | *didactylus* | Sítio Ananias (25°47′08″S, 48°43′03″W), municipality of Guaratuba | PR | 25 | *Pie et al.* (*2013*; as "*Brachycephalus* sp. nov. 1") |
| *Brachycephalus* sp. 1 | Southern Serra do Mar | *didactylus* | Truticultura (26°01′33″S, 48°52′02″W), municipality of Garuva | SC | 90 | *Pie et al.* (*2013*; as "*Brachycephalus* sp. nov. 1") |
| *B. alipioi* | Northern Serra da Mantiqueira | *ephippium* | Fazenda Aoki or Fazenda dos Japoneses (20°28′24″S, 41°00′36″W), boundary of the municipalities of Vargem Alta and Domingos Martins | ES | 1,070–1,100 | *Pombal Jr & Gasparini (2006)*, *Clemente-Carvalho et al. (2009)*, *Clemente-Carvalho et al. (2011b)*, *Clemente-Carvalho et al. (2012)*, *Pombal Jr & Izecksohn (2011)*, *Pie et al. (2013)* |
| *B. alipioi* | Northern Serra da Mantiqueira | *ephippium* | Forno Grande (20°31′41″S, 41°06′51″W), Parque Estadual de Forno Grande, municipality of Castelo | ES | ? (1,430?) | *Pie et al. (2013)* |
| *B. bufonoides* | Northern Serra do Mar | *ephippium* | Serra de Macaé (22°18′02″S, 42°18′20″W), municipality of Nova Friburgo | RJ | ? (1,100?) | *Miranda-Ribeiro (1920)*, *Pombal Jr (2010)*, *Pombal Jr & Izecksohn (2011)* |
| *B. crispus* | Central Serra do Mar | *ephippium* | Bacia B, Núcleo Cunha, Parque Estadual da Serra do Mar (23°15′15″S, 45°01′58″W), municipality of Cunha | SP | 800-1,100 | *Condez et al. (2014)* |
| *B. ephippium* | Northern Serra da Mantiqueira | *ephippium* | Serra do Pai Inácio (20°48′13″S, 42°29′07″W), Parque Estadual da Serra do Brigadeiro, boundary of the municipalities of Ervália and Miradouro | MG | ? (1,350?) | *Pombal Jr & Izecksohn (2011)* |
| *B. ephippium* | Southern Serra da Mantiqueira | *ephippium* | Condomínio Ermida (23°14′13″S, 46°58′52″W), Serra do Japi, municipality of Jundiaí | SP | 1,225 | *Pie et al. (2013)* |
| *B. ephippium* | Southern Serra da Mantiqueira | *ephippium* | Hotel Fazenda Pé da Serra (22°51′56″S, 45°31′40″W), municipality of Pindamonhangaba | SP | 700 | *Pie et al. (2013)* |
| *B. ephippium* | Southern Serra da Mantiqueira | *ephippium* | Lago Azul (22°27′23″S, 44°36′34″W), Parque Nacional do Itatiaia, municipality of Itatiaia | RJ | 750 | *Pie et al. (2013)* |

**Table 1** (*continued*)

| Species[a] | Relief units[b] | Group[c] | Locality[d] | State | Altitude | Source[e] |
|---|---|---|---|---|---|---|
| *B. ephippium* | Southern Serra da Mantiqueira | *ephippium* | Maromba (22°25′43″S, 44°37′11″W), Parque Nacional do Itatiaia, municipality of Itatiaia | RJ | 1,125 | *Pie et al. (2013)* |
| *B. ephippium* | Southern Serra da Mantiqueira | *ephippium* | Monteiro Lobato (22°57′07″S, 45°50′20″W), municipality of Monteiro Lobato | SP | 700 | *Pombal Jr & Izecksohn (2011)* |
| *B. ephippium* | Southern Serra da Mantiqueira | *ephippium* | Observatório de Capricórnio (22°53′54″S, 46°49′01″W), Serra das Cabras, Joaquim Egídio District, boundary of the municipalities of Campinas and Morungaba | SP | 1,085 | *Pombal Jr (1999)*, *Pombal Jr, Sazima & Haddad (1994)*, *Pombal Jr & Izecksohn (2011)*, *Pie et al. (2013)* |
| *B. ephippium* | Southern Serra da Mantiqueira | *ephippium* | Parque Municipal de Itapetinga (Grota Funda) (23°11′07″S, 46°31′47″W), municipality of Atibaia | SP | 900–1,250 | *Giaretta (1999)*, *Giaretta et al. (1999)*, *Clemente-Carvalho et al. (2009)*, *Clemente-Carvalho et al. (2011b)*, *Pie et al. (2013)* |
| *B. ephippium* | Southern Serra da Mantiqueira | *ephippium* | Reserva Biológica da Serra do Japi (23°17′07″S, 47°00′05″W), Serra do Japi, boundary of the municipalities of Jundiaí and Cabreúva | SP | 1,000 | *Giaretta et al. (1997)*, *Pombal Jr & Izecksohn (2011)*, *Clemente-Carvalho et al. (2009)*, *Clemente-Carvalho et al. (2016)*, *Pie et al. (2013)* |
| *B. ephippium* | Southern Serra da Mantiqueira | *ephippium* | Reserva Ecológica do Trabiju (22°48′01″S, 45°32′03″W), Trabiju, municipality of Pindamonhangaba | SP | ? (1,000?) | *Pombal Jr & Izecksohn (2011)* |
| *B. ephippium* | Southern Serra da Mantiqueira | *ephippium* | Reserva Pedra Branca (22°56′22″S, 45°41′04″W), municipality of Tremembé | SP | ? (890?) | *Pombal Jr & Izecksohn (2011)* |
| *B. ephippium* | Southern Serra da Mantiqueira | *ephippium* | Santo Antônio do Pinhal (22°49′28″S, 45°40′20″W), municipality of Santo Antônio do Pinhal | SP | 1,080 | *Pombal Jr & Izecksohn (2011)* |
| *B. ephippium* | Southern Serra da Mantiqueira | *ephippium* | São Francisco Xavier (22°53′44″S, 45°58′04″W), municipality of São José dos Campos | SP | 1,000 | *Clemente-Carvalho et al. (2008)*, *Clemente-Carvalho et al. (2011a)*, *Clemente-Carvalho et al. (2016)*, *Pombal Jr & Izecksohn (2011)*, *Pie et al. (2013)* |
| *B. ephippium* | Southern Serra da Mantiqueira | *ephippium* | Serra Negra (21°57′28″S, 43°47′20″W), municipality of Santa Bárbara do Monte Verde | MG | ? | *Campos (2011;* as "BMV MG2") |
| *B. ephippium* | Paraíba do Sul | *ephippium* | Serra da Concórdia (22°20′30″S, 43°44′04″W), Parque Estadual Serra da Concórdia, Barão de Juparanã, municipality of Valença | RJ | ? (900?) | *Pombal Jr & Izecksohn (2011)* |
| *B. ephippium* | Northern Serra do Mar | *ephippium* | Alto do Soberbo (22°27′15″S, 42°59′21″W), municipality of Teresópolis | RJ | 1,250 | *Pombal Jr & Izecksohn (2011)* |

**Table 1** (*continued*)

| Species[a] | Relief units[b] | Group[c] | Locality[d] | State | Altitude | Source [e] |
|---|---|---|---|---|---|---|
| *B. ephippium* | Northern Serra do Mar | *ephippium* | Comary (22°27′22″S, 42°58′24″W), municipality of Teresópolis | RJ | 990 | *Pombal Jr & Izecksohn (2011)* |
| *B. ephippium* | Northern Serra do Mar | *ephippium* | Floresta dos Macacos (22°58′15″S, 43°15′24″W), municipality of Rio de Janeiro | RJ | ? (450?) | *Pombal Jr & Izecksohn (2011)* |
| *B. ephippium* | Northern Serra do Mar | *ephippium* | Garrafão (22°28′04″S, 43°01′52″W), municipality of Guapimirim | RJ | ? (1,785?) | *Pombal Jr & Izecksohn (2011)* |
| *B. ephippium* | Northern Serra do Mar | *ephippium* | Pedra Branca (22°55′55″S, 43°28′23″W), Serra da Pedra Branca, municipality of Rio de Janeiro | RJ | 1,000 | *Pombal Jr (2001)*, *Pombal Jr & Izecksohn (2011)* |
| *B. ephippium* | Northern Serra do Mar | *ephippium* | Represa do Rio Grande (22°55′58″S, 43°26′36″W), Parque Estadual da Pedra Branca, municipality of Rio de Janeiro | RJ | ? (150?) | *Pombal Jr & Izecksohn (2011)*, *Pie et al. (2013)* |
| *B. ephippium* | Northern Serra do Mar | *ephippium* | Reserva Ecológica Rio das Pedras (22°59′00″S, 44°06′45″W), municipality of Mangaratiba | RJ | 200-1,110 | *Carvalho-e-Silva, Silva & Carvalho-e-Silva (2008)* |
| *B. ephippium* | Northern Serra do Mar | *ephippium* | Riacho Beija-flor (22°27′04″S, 43°00′04″W), Parque Nacional da Serra dos Órgãos, municipality of Teresópolis | RJ | 1,195 | *Pie et al. (2013)* |
| *B. ephippium* | Northern Serra do Mar | *ephippium* | Rocio District (22°28′23″S, 43°14′38″W), municipality of Petrópolis | RJ | 950 | *Pie et al. (2013)* |
| *B. ephippium* | Northern Serra do Mar | *ephippium* | Serra do Tinguá (22°35′31″S, 43°28′16″W), municipality of Nova Iguaçu | RJ | ? (950?) | *Pombal Jr & Izecksohn (2011)* |
| *B. ephippium* | Northern Serra do Mar | *ephippium* | Vale da Revolta (22°26′17″S, 42°56′19″W), municipality of Teresópolis | RJ | 1,035 | *Pombal Jr & Izecksohn (2011)* |
| *B. ephippium* | Northern Serra do Mar | *ephippium* | Varginha (22°24′34″S, 42°52′11″W), municipality of Teresópolis | RJ | ? (825?) | *Pombal Jr & Izecksohn (2011)* |
| *B. ephippium* | Central Serra do Mar | *ephippium* | Bonito (22°42′51″S, 44°34′39″W), Serra da Bocaina, municipality of São José do Barreiro | SP | ? (1,660?) | *Pombal Jr & Izecksohn (2011)* |
| *B. ephippium* | Central Serra do Mar | *ephippium* | Estação Ecológica de Bananal (22°48′05″S, 44°22′12″W), Serra da Bocaina, municipality of Bananal | SP | ? (1,200?) | *Zaher, Aguiar & Pombal (2005)* |
| *B. ephippium* | Central Serra do Mar | *ephippium* | Lídice District (22°50′01″S, 44°11′32″W), municipality of Rio Claro | RJ | ? (650?) | *Pombal Jr (2001)*, *Pombal Jr & Izecksohn (2011)* |
| *B. ephippium* | Central Serra do Mar | *ephippium* | Pedra Branca (23°10′38″S, 44°47′19″W), Serra da Bocaina, municipality of Parati | RJ | ? (630?) | *Pombal Jr (2001)*, *Pombal Jr & Izecksohn (2011)* |
| *B. ephippium* | Central Serra do Mar | *ephippium* | Península do Bororé (23°47′11″S, 46°38′45″W), Represa Billings, Grajaú District, municipality of São Paulo | SP | 780 | *Pie et al. (2013*; as "*Brachycephalus nodoterga*"), *Abbeg et al. (2015*; as "clearly refers to another species (than *B. nodoterga* of *Pie et al. 2013*)") |

Bornschein et al. (2016), *PeerJ*, DOI 10.7717/peerj.2490

| Species[a] | Relief units[b] | Group[c] | Locality[d] | State | Altitude | Source [e] |
|---|---|---|---|---|---|---|
| *B. ephippium* | Central Serra do Mar | *ephippium* | Reserva Florestal de Morro Grande (23°42′08″S, 46°58′22″W), municipality of Cotia | SP | ? (990?) | *Dixo & Verdade (2006)* |
| *B. garbeanus* | Northern Serra do Mar | *ephippium* | Alto Caledônia (22°20′10″S, 42°33′20″W), municipality of Nova Friburgo | RJ | ? (1,070?) | *Pombal Jr & Izecksohn (2011)* |
| *B. garbeanus* | Northern Serra do Mar | *ephippium* | Baixo Caledônia (22°21′33″S, 42°34′12″W), municipality of Nova Friburgo | RJ | 1,600–1,900 | *Pombal Jr & Izecksohn (2011), Siqueira et al. (2011), Siqueira, Vrcibradic & Rocha (2013), Dorigo et al. (2012)* |
| *B. garbeanus* | Northern Serra do Mar | *ephippium* | Macaé de Cima (22°21′37″S, 42°17′50″W), municipality of Nova Friburgo | RJ | 1,130 | *Clemente-Carvalho et al. (2008; as "B. ephippium") Clemente-Carvalho et al. (2009; as "B. ephippium"), Clemente-Carvalho et al. (2011b), Clemente-Carvalho et al. (2011a; "b", as "B. ephippium"), Pombal Jr & Izecksohn (2011), Pie et al. (2013)* |
| *B. garbeanus* | Northern Serra do Mar | *ephippium* | Morro São João (22°22′47″S, 42°30′34″W), municipality of Nova Friburgo | RJ | ? (1,550?) | *Pombal Jr & Izecksohn (2011)* |
| *B. garbeanus* | Northern Serra do Mar | *ephippium* | Serra de Macaé (22°18′02″S, 42°18′20″W), municipality of Nova Friburgo | RJ | ? (1,100?) | *Miranda-Ribeiro (1920), Pombal Jr (2010), Pombal Jr & Izecksohn (2011)* |
| *B. garbeanus* | Northern Serra do Mar | *ephippium* | Serra Nevada (22°21′46″S, 42°32′48″W), municipality of Nova Friburgo | RJ | 1,190 | *Pombal Jr & Izecksohn (2011)* |
| *B. garbeanus* | Northern Serra do Mar | *ephippium* | Theodoro de Oliveira (second position: 22°21′48″S, 42°33′13″W), Parque Estadual dos Três Picos, municipality of Nova Friburgo | RJ | 1,400 | *Pombal Jr & Izecksohn (2011), Siqueira et al. (2011), Siqueira, Vrcibradic & Rocha (2013)* |
| *B. guarani* | Central Serra do Mar | *ephippium* | Morro Prumirim (23°20′50″S, 45°01′37″W), municipality of Ubatuba | SP | 500–900 | *Giaretta (1999; altitude determined by Clemente-Carvalho et al. (2012)), Clemente-Carvalho et al. (2012), Condez et al. (2014)* |
| *B. margarita-tus* | Northern Serra do Mar | *ephippium* | Castelo Country Club (22°32′21″S, 43°13′08″W), municipality of Petrópolis | RJ | 980 | *Pombal Jr & Izecksohn (2011)* |
| *B. margarita-tus* | Northern Serra do Mar | *ephippium* | Castelo Montebello (22°24′24″S, 42°58′06″W), municipality of Teresópolis | RJ | 920 | *Pombal Jr & Izecksohn (2011)* |
| *B. margarita-tus* | Northern Serra do Mar | *ephippium* | Independência (22°32′58″S, 43°12′27″W), municipality of Petrópolis | RJ | 860 | *Pombal Jr & Izecksohn (2011)* |
| *B. margarita-tus* | Northern Serra do Mar | *ephippium* | Morro Azul (22°28′34″S, 43°34′40″W), municipality of Engenheiro Paulo de Frontin | RJ | 620 | *Campos (2011; as "BPF RJ2"), Pombal Jr & Izecksohn (2011)* |

Peer**J**

**Table 1** (*continued*)

| Species[a] | Relief units[b] | Group[c] | Locality[d] | State | Altitude | Source[e] |
|---|---|---|---|---|---|---|
| *B. margarita-tus* | Northern Serra do Mar | *ephippium* | Quitandinha (22°31′47″S, 43°12′26″W), municipality of Petrópolis | RJ | 925 | *Pombal Jr & Izecksohn (2011)* |
| *B. margarita-tus* | Northern Serra do Mar | *ephippium* | Sacra Família do Tinguá (22°29′11″S, 43°36′18″W), municipality of Engenheiro Paulo de Frontin | RJ | 600 | *Izecksohn (1971*; as "*B. ephippium*"), *Pombal Jr (2001*; as "*Brachycephalus* cf. *ephippium*"), *Pombal Jr & Izecksohn (2011)*, *Pie et al. (2013*; as "*B. ephippium*") |
| *B. nodoterga* | Central Serra do Mar | *ephippium* | Estação Biológica de Boracéia (second position: 23°38′00″S, 45°52′00″W), municipality of Salesópolis | SP | 945 | *Pombal Jr, Wistuba & Bornschein (1998)*, *Pombal Jr (2001)*, *Pombal Jr (2010)*, *Ribeiro, Alves & Haddad (2005)*, *Alves et al. (2006)*, *Alves et al. (2009)*, *Haddad et al. (2010)*, *Pombal Jr & Izecksohn (2011)*, *Pie et al. (2013)*, *Abegg et al. (2015)*, *Clemente-Carvalho et al. (2016)* |
| *B. nodoterga* | Central Serra do Mar | *ephippium* | Fazenda Paiva Ramos (23°28′21″S, 46°47′25″W), Serra de São Roque, municipality of Osasco | SP | 820 | *Abegg et al. (2015)* |
| *B. nodoterga* | Central Serra do Mar | *ephippium* | Pico do Ramalho (23°51′42″S, 45°21′28″W), Ilha de São Sebastião, municipality of Ilhabela | SP | 700–900 | *Ribeiro (2006*; as "*Brachycephalus* sp. aff. *nodoterga*"), *Pombal Jr & Izecksohn (2011)*, *Pie et al. (2013)*, *Abegg et al. (2015)*, *Clemente-Carvalho et al. (2016)* |
| *B. nodoterga* | Central Serra do Mar | *ephippium* | Serra da Cantareira (23°27′13″S, 46°38′11″W), Parque Estadual da Cantareira, municipality of São Paulo | SP | ? (850?) | *Miranda-Ribeiro (1920)*, *Ribeiro, Alves & Haddad (2005)*, *Alves et al. (2006)*, *Alves et al. (2009)*, *Clemente-Carvalho et al. (2009)*, *Clemente-Carvalho et al. (2011b)*, *Clemente-Carvalho et al. (2012)*, *Clemente-Carvalho et al. (2016)*, *Haddad et al. (2010)*, *Pombal Jr (2010)*, *Pombal Jr & Izecksohn (2011)*, *Condez et al. (2014)*, *Abegg et al. (2015)* |
| *B. pitanga* | Central Serra do Mar | *ephippium* | Fazenda Capricórnio (23°22′36″S, 45°04′07″W), municipality of Ubatuba | SP | ? (450?) | *Alves et al. (2009)*, *Campos, Silva & Sebben (2010*; as "*Brachycephalus* sp. 2"), *Campos (2011)*, *Pie et al. (2013)* |
| *B. pitanga* | Central Serra do Mar | *ephippium* | Núcleo Santa Virgínia (23°19′23″S, 45°05′19″W), Parque Estadual da Serra do Mar, municipality of São Luis do Paraitinga | SP | 980–1,140 | *Oliveira (2013)*, *Tandel et al. (2014)*, *Oliveira & Haddad (2015)* |
| *B. pitanga* | Central Serra do Mar | *ephippium* | SP 125 - municipality of São Luís do Paraitinga (23°22′57″S, 45°09′59″W) | SP | 935–950 | This study, DZUP |

**Table 1** (*continued*)

| Species[a] | Relief units[b] | Group[c] | Locality[d] | State | Altitude | Source [e] |
|---|---|---|---|---|---|---|
| *B. pitanga* | Central Serra do Mar | *ephippium* | Trilha do Ipiranga 50 m from the Rio Ipiranga (23°20′39″S, 45°08′16″W), Núcleo Santa Virgínia, Parque Estadual da Serra do Mar, municipality of São Luis do Paraitinga | SP | 900–960 | *Alves et al. (2009)*, *Clemente-Carvalho et al. (2009)*, *Clemente-Carvalho et al. (2011b)*, *Araújo et al. (2012)*, *Oliveira (2013)*, *Pie et al. (2013)* |
| *B. toby* | Central Serra do Mar | *ephippium* | Morro do Corcovado (23°27′20″S, 45°11′49″W), Parque Estadual da Serra do Mar, municipality of Ubatuba | SP | 750 | *Haddad et al. (2010)*, *Clemente-Carvalho et al. (2011b)*, *Clemente-Carvalho et al. (2012)*, *Pie et al. (2013)*, *Condez et al. (2014)* |
| *B. vertebralis* | Central Serra do Mar | *ephippium* | Morro Cuzcuzeiro (23°18′03″S, 44°47′30″W), Núcleo Picinguaba, Parque Estadual da Serra do Mar, municipality of Ubatuba | SP | 900 | *Clemente-Carvalho et al. (2011b)*, *Pie et al. (2013)*, *Condez et al. (2014)* |
| *B. vertebralis* | Central Serra do Mar | *ephippium* | Pedra Branca (23°10′38″S, 44°47′19″W), Serra da Bocaina, municipality of Parati | RJ | ? (630?) | *Pombal Jr (2001)*, *Pombal Jr (2010)*, *Clemente-Carvalho et al. (2009)*, *Pie et al. (2013)* |
| *Brachycephalus* sp. 2 | Northern Serra do Mar | *ephippium* | Theodoro de Oliveira (first position: 22°22′11″S, 42°33′25″W), Parque Estadual dos Três Picos, municipality of Nova Friburgo | RJ | 1,100–1,200 | *Siqueira et al. (2011*; as "*Brachycephalus* sp."), (*Siqueira, Vrcibradic & Rocha, 2013*; as "*Brachycephalus* sp. nov.") |
| *Brachycephalus* sp. 3 | Central Serra do Mar | *ephippium* | Paranapiacaba (23°46′30″S, 46°17′57″W), municipality of Santo André | SP | 825 | *Pombal Jr & Izecksohn* (*2011*; as "*B. ephippium*"), *Pie et al.* (*2013*; as "*Brachycephalus* sp. 1" ) |
| *Brachycephalus* sp. 3 | Central Serra do Mar | *ephippium* | Parque Natural Municipal Nascentes de Paranapiacaba (23°46′10″S, 46°17′36″W), municipality of Santo André | SP | ? (800–1,164?) | *Trevine et al.* (*2014*; as "*Brachycephalus* sp.") |
| *Brachycephalus* sp. 3 | Central Serra do Mar | *ephippium* | Reserva Biológica do Alto da Serra de Paranapiacaba (23°46′40″S, 46°18′45″W), municipality of Santo André | SP | 800 | *Pie et al.* (*2013*; as "*Brachycephalus* sp. 1") |
| *B. auroguttatus* | Southern Serra do Mar | *pernix* | Pedra da Tartaruga (26°00′21″S, 48°55′25″W), municipality of Garuva | SC | 1,070–1,100 | This study, DZUP, *Firkowski* (*2013*; without species identification), *Ribeiro et al. (2015)* |
| *B. boticario* | Leste Catarinense | *pernix* | Morro do Cachorro (26°46′42″S, 49°01′57″W), boundary of the municipalities of Blumenau, Gaspar, and Luiz Alves | SC | 755–795 | This study, DZUP, *Firkowski* (*2013*; without species identification), *Ribeiro et al. (2015)* |
| *B. brunneus* | Southern Serra do Mar | *pernix* | Camapuã (25°15′59″S, 48°50′16″W), Serra dos Órgãos, boundary of the municipalities of Campina Grande do Sul and Antonina | PR | 1,595 | *Fontoura, Ribeiro & Pie (2011)*, *Firkowski* (*2013*; without species identification), *Pie et al. (2013)* |

Bornschein et al. (2016), *PeerJ*, DOI 10.7717/peerj.2490

| Species[a] | Relief units[b] | Group[c] | Locality[d] | State | Altitude | Source [e] |
|---|---|---|---|---|---|---|
| *B. brunneus* | Southern Serra do Mar | *pernix* | Caranguejeira (25°20′27″S, 48°54′31″W), Serra da Graciosa, municipality of Quatro Barras | PR | 1,095–1,110 | This study, DZUP, *Firkowski* (*2013*; without species identification) |
| *B. brunneus* | Southern Serra do Mar | *pernix* | Caratuva (25°14′33″S, 48°50′04″W), Serra dos Órgãos, municipality of Campina Grande do Sul | PR | 1,300–1,630 | *Ribeiro, Alves & Haddad (2005)*, *Clemente-Carvalho et al. (2009)*, *Clemente-Carvalho et al. (2011b)*, *Campos (2011)*, *Fontoura, Ribeiro & Pie (2011)*, *Pombal Jr & Izecksohn* (*2011*; including "Pico Paraná"), *Firkowski* (*2013*; without species identification; including "Pico Paraná"), *Pie et al. (2013)* |
| *B. brunneus* | Southern Serra do Mar | *pernix* | Getúlio (25°14′18″S, 48°50′13″W), Serra dos Órgãos, municipality of Campina Grande do Sul | PR | 1,450–1,490 | This study, *Pie et al. (2013)* |
| *B. brunneus* | Southern Serra do Mar | *pernix* | Mãe Catira (25°20′51″S, 48°54′25″W), Serra da Graciosa, municipality of Quatro Barras | PR | 1,135–1,405 | *Firkowski* (*2013*; without species identification), *Pie et al. (2013*; as "*Brachycephalus* sp. nov. 2") |
| *B. brunneus* | Southern Serra do Mar | *pernix* | Tupipiá (25°14′31″S, 48°47′47″W), Serra dos Órgãos, municipality of Antonina | PR | 1,260 | *Firkowski* (*2013*; without species identification), *Pie et al. (2013)* |
| *B. ferruginus* | Southern Serra do Mar | *pernix* | Olimpo (25°27′03″S, 48°54′59″W), Serra do Marumbi, municipality of Morretes | PR | 965–1,470 | *Alves et al. (2006)*, *Clemente-Carvalho et al. (2009)*, *Clemente-Carvalho et al. (2011b)*, *Pombal Jr & Izecksohn (2011)*, *Firkowski* (*2013*; without species identification), *Pie et al. (2013)* |
| *B. fuscolineatus* | Leste Catarinense | *pernix* | Morro do Baú (26°47′58″S, 48°55′47″W), municipality of Ilhota | SC | 640–790 | *Firkowski* (*2013*; without species identification), *Pie et al. (2013*; as "*Brachycephalus* sp. nov. 9"), *Ribeiro et al. (2015)* |
| *B. izecksohni* | Southern Serra do Mar | *pernix* | Torre da Prata, Serra da Prata (25°37′25″S, 48°41′31″W), boundary of the municipalities of Morretes, Paranaguá, and Guaratuba | PR | 980–1,340 | *Ribeiro, Alves & Haddad (2005)*, *Clemente-Carvalho et al. (2009)*, *Clemente-Carvalho et al. (2011b)*, *Pombal Jr & Izecksohn (2011)*, *Firkowski* (*2013*; without species identification), *Pie et al. (2013)* |
| *B. leopardus* | Southern Serra do Mar | *pernix* | Morro dos Perdidos (25°53′22″S, 48°57′22″W), municipality of Guaratuba | PR | 1,400–1,420 | *Firkowski* (*2013*; without species identification), *Pie et al. (2013*; as "*Brachycephalus* sp. nov. 4") |
| *B. leopardus* | Southern Serra do Mar | *pernix* | Serra do Araçatuba (25°54′07″S, 48°59′47″W), municipality of Tijucas do Sul | PR | 1,640 | *Firkowski* (*2013*; without species identification), *Pie et al. (2013*; as "*Brachycephalus* sp. nov. 4"), *Ribeiro et al. (2015)* |
| *B. mariaeterezae* | Southern Serra do Mar | *pernix* | Reserva Particular do Patrimônio Natural Caetezal, top of the Serra Queimada (26°06′51″S, 49°03′45″W), municipality of Joinville | SC | 1,265–1,270 | *Firkowski* (*2013*; without species identification), *Pie et al. (2013*; as "*Brachycephalus* sp. nov. 6"), *Ribeiro et al. (2015)* |

**Table 1** (*continued*)

| Species[a] | Relief units[b] | Group[c] | Locality[d] | State | Altitude | Source[e] |
|---|---|---|---|---|---|---|
| *B. olivaceus* | Southern Serra do Mar | *pernix* | Castelo dos Bugres (second position: 26°13′59″S, 49°03′13″W), municipality of Joinville | SC | 800–835 | *Firkowski* (*2013*; without species identification), *Pie et al.* (*2013*; as "*Brachycephalus* sp. nov. 7"), *Ribeiro et al.* (*2015*) |
| *B. olivaceus* | Southern Serra do Mar | *pernix* | Base of the Serra Queimada (26°04′57″S, 49°03′59″W), municipality of Joinville | SC | 985 | *Pie et al.* (*2013*; as "*Brachycephalus* sp. nov. 7"), *Ribeiro et al.* (*2015*) |
| *B. olivaceus* | Leste Catari-nense | *Pernix* | Morro do Boi (26°24′42″S, 49°12′59″W), municipality of Corupá | SC | 690–920 | This study, MHNCI, *Pie et al.* (*2013*; as "*Brachycephalus* sp. 3") |
| *B. pernix* | Southern Serra do Mar | *pernix* | Anhangava (25°23′19″S, 49°00′15″W), Serra da Baitaca, municipality of Quatro Barras | PR | 1,135–1,405 | *Pombal Jr, Wistuba & Bornschein* (*1998*), *Wistuba* (*1998*), *Pires Jr et al.* (*2005*), *Silva, Campos & Sebben* (*2007*), *Clemente-Carvalho et al.* (*2009*), *Clemente-Carvalho et al.* (*2011b*), *Campos, Silva & Sebben* (*2010*), *Campos* (*2011*), *Pombal Jr & Izecksohn* (*2011*), *Firkowski* (*2013*; without species identification), *Pie et al.* (*2013*), *Ribeiro et al.* (*2014*) |
| *B. pombali* | Southern Serra do Mar | *pernix* | Morro dos Padres (25°36′40″S, 48°51′22″W), Serra da Igreja, municipality of Morretes | PR | 1,060–1,300 | *Alves et al.* (*2006*), *Clemente-Carvalho et al.* (*2009*), *Clemente-Carvalho et al.* (*2011b*), *Firkowski* (*2013*; without species identification), *Pie et al.* (*2013*) |
| *B. pombali* | Southern Serra do Mar | *pernix* | Trail to Morro dos Padres (25°35′58″S, 48°51′57″W), municipality of Morretes | PR | 845–1,060 | *Pie et al.* (*2013*) |
| *B. quiririensis* | Southern Serra do Mar | *pernix* | Serra do Quiriri (26°01′17″S, 48°59′47″W), municipality of Campo Alegre | SC | 1,240–1,318 | *Firkowski* (*2013*; without species identification), *Pie et al.* (*2013*; as "*Brachycephalus* sp. nov. 5"), *Pie & Ribeiro* (*2015*) |
| *B. quiririensis* | Southern Serra do Mar | *pernix* | Serra do Quiriri (26°01′42″S, 48°57′11″W), municipality of Garuva | SC | 1,320–1,380 | *Pie et al.* (*2013*; as "*Brachycephalus* sp. nov. 5"), *Pie & Ribeiro* (*2015*) |
| *B. tridactylus* | Southern Serra do Mar | *pernix* | Serra do Morato (25°08′09″S, 48°17′59″W), Reserva Natural Salto Morato, municipality of Guaraqueçaba | PR | 805–910 | This study, DZUP, *Garey et al.* (*2012*), *Bornschein et al.* (*2015*) |
| *B. verrucosus* | Southern Serra do Mar | *pernix* | Morro da Tromba (26°12′44″S, 48°57′29″W), municipality of Joinville | SC | 455–945 | *Firkowski* (*2013*; without species identification), *Pie et al.* (*2013*; as "*Brachycephalus* sp. nov. 8"), *Ribeiro et al.* (*2015*) |
| *Brachycephalus* sp. 4 | Southern Serra do Mar | *pernix* | Morro do Canal (25°30′55″S, 48°58′56″W), municipality of Piraquara | PR | 1,320 | DZUP, *Firkowski* (*2013*; without species identification) |
| *Brachycephalus* sp. 4 | Southern Serra do Mar | *pernix* | Morro do Vigia (25°30′33″S, 48°58′58″W), municipality of Piraquara | PR | 1,250 | *Firkowski* (*2013*; without species identification), *Pie et al.* (*2013*; as "*Brachycephalus* sp. nov. 3") |
| *Brachycephalus* sp. 5 | Southern Serra do Mar | *pernix* | Pedra Branca do Araraquara (25°56′00″S, 48°52′50″W), Serra do Araraquara, municipality of Guaratuba | PR | 1,000 | This study, DZUP |

Peer**J**

**Table 1** (*continued*)

| Species[a] | Relief units[b] | Group[c] | Locality[d] | State | Altitude | Source[e] |
|---|---|---|---|---|---|---|
| *Brachycephalus* sp. 5 | Southern Serra do Mar | *pernix* | Serra Canasvieiras (25°36′58″S, 48°46′59″W), boundary of the municipalities of Guaratuba and Morretes | PR | 1,080 | This study, DZUP, *Firkowski* (*2013*; without species identification) |
| *Brachycephalus* sp. 6 | Southern Serra do Mar | *pernix* | Serra do Salto (25°42′07″S, 49°03′44″W), Malhada District, municipality of São José dos Pinhais | PR | 1,095–1,160 | This study, DZUP, *Firkowski* (*2013*; without species identification), *Pie et al.* (*2013*; as "*Brachycephalus* sp. 2") |
| Excluded records[f] | | | | | | |
| *B. hermogenesi* | Central Serra do Mar | *didactylus* | Ubatuba | SP | ? | *Silva, Campos & Sebben (2007)*, *Campos, Silva & Sebben (2010)*, *Campos (2011)* |
| *B. hermogenesi* | Paranapiacaba | *didactylus* | Municipality of Piedade | SP | ? | *Condez, Sawaya & Dixo (2009)*, *Clemente-Carvalho et al. (2011b)* |
| *B. alipioi* | Northern Serra da Mantiqueira | *ephippium* | Santa Teresa | ES | ? | *Pombal Jr & Gasparini (2006)*, *Pombal Jr & Izecksohn (2011)* |
| *B.* cf. *crispus* | Central Serra do Mar | *ephippium* | Cunha | SP | ? | *Campos, Silva & Sebben* (*2010*; as "*Brachycephalus* sp. 3"), *Campos* (*2011*; as "BCU SP2") |
| *B. ephippium* | Southern Serra da Mantiqueira | *ephippium* | Atibaia | SP | ? | *Pires Jr et al. (2003)*, *Pires Jr et al. (2005)*, *Ananias, Giaretta & Recco-Pimentel (2006)*, *Clemente-Carvalho et al. (2008)*, *Clemente-Carvalho et al. (2009)*, *Clemente-Carvalho et al. (2011b)*, *Clemente-Carvalho et al. (2011a)*, *Clemente-Carvalho et al. (2016)*, *Campos, Silva & Sebben (2010)*, *Campos* (*2011*; as "BAT SP3"), *Pombal Jr & Izecksohn (2011)* |
| *B. ephippium* | Southern Serra da Mantiqueira | *ephippium* | Itamonte | MG | ? | *Silva, Campos & Sebben (2007)*, *Campos, Silva & Sebben (2010)*, *Campos* (*2011*; as "BIT MG1") |
| *B. ephippium* | Southern Serra da Mantiqueira | *ephippium* | Itatiaia | RJ | ? | *Pombal Jr & Izecksohn (2011)* |
| *B. ephippium* | Southern Serra da Mantiqueira | *ephippium* | Joaquim Egídio | SP | ? | *Clemente-Carvalho et al. (2016)* |
| *B. ephippium* | Southern Serra da Mantiqueira | *ephippium* | Jundiaí | SP | ? | *Miranda-Ribeiro (1920)*, *Clemente-Carvalho et al. (2008)*, *Clemente-Carvalho et al. (2011a)* |
| *B. ephippium* | Southern Serra da Mantiqueira | *ephippium* | Piquete | SP | ? | *Miranda-Ribeiro (1920)*, *Pombal Jr & Izecksohn (2011)* |

**Table 1** (*continued*)

| Species[a] | Relief units[b] | Group[c] | Locality[d] | State | Altitude | Source[e] |
|---|---|---|---|---|---|---|
| *B. ephippium* | Northern Serra do Mar | *ephippium* | Angra dos Reis | RJ | ? | *Pombal Jr & Izecksohn (2011)* |
| *B. ephippium* | Northern Serra do Mar | *ephippium* | Floresta da Tijuca, municipality of Rio de Janeiro | RJ | ? | *Pombal Jr & Izecksohn (2011)* |
| *B. ephippium* | Northern Serra do Mar | *ephippium* | Guapiaçu (22°26′34″S, 42°45′38″W), municipality of Cachoeira de Macacu | RJ | 35 | *Pombal Jr & Izecksohn (2011)* |
| *B. ephippium* | Northern Serra do Mar | *ephippium* | Mangaratiba | RJ | ? | *Campos (2011; as "BMA RJ1"), Pombal Jr & Izecksohn (2011)* |
| *B. ephippium* | Northern Serra do Mar | *ephippium* | Parque Nacional da Serra dos Órgãos | RJ | ? | *Pombal Jr & Izecksohn (2011)* |
| *B. ephippium* | Northern Serra do Mar | *ephippium* | Rio de Janeiro | RJ | ? | *Pombal Jr & Izecksohn (2011)* |
| *B. ephippium* | Northern Serra do Mar | *ephippium* | Teresópolis | RJ | ? | *Pires Jr et al. (2002), Pires Jr et al. (2005), Silva, Campos & Sebben (2007), Campos, Silva & Sebben (2010), Campos (2011; as "BTE RJ4"), Pombal Jr & Izecksohn (2011)* |
| *B. ephippium* | Central Serra do Mar | *ephippium* | Bocaina | SP | ? | *Pombal Jr & Izecksohn (2011)* |
| *B. ephippium* | Central Serra do Mar | *ephippium* | Cotia | SP | ? | *Silva, Campos & Sebben (2007), Campos, Silva & Sebben (2010), Campos (2011; as "BCO SP5")* |
| *B. ephippium* | Central Serra do Mar | *ephippium* | Fazenda Papagaio, Serra da Bocaina | SP | ? | *Pombal Jr & Izecksohn (2011)* |
| *B. ephippium* | Central Serra do Mar | *ephippium* | Mogi das Cruzes | SP | ? | *Pires Jr et al. (2005), Silva, Campos & Sebben (2007), Campos, Silva & Sebben (2010), Campos (2011; as "BMC SP4"), Pombal Jr & Izecksohn (2011)* |
| *B. ephippium* | Central Serra do Mar | *ephippium* | Serra da Bocaina | SP/RJ | ? | *Pombal Jr & Izecksohn (2011)* |
| *B. garbeanus* | Northern Serra do Mar | *ephippium* | Nova Friburgo | RJ | ? | *Silva, Campos & Sebben (2007; as "B. ephippium"), Campos, Silva & Sebben (2010; as "B. ephippium"), Campos (2011)* |
| *B. margaritatus* | Northern Serra do Mar | *ephippium* | Sítio do Pau Ferro, Morro Azul, municipality of Engenheiro Paulo de Frontin | RJ | ? | *Pombal Jr & Izecksohn (2011)* |
| *B.* cf. *margaritatus* | Northern Serra do Mar | *ephippium* | Petrópolis | RJ | ? | *Campos (2011; as "BPT RJ3")* |
| *B. nodoterga* | Central Serra do Mar | *ephippium* | Salesópolis | SP | ? | *Pires Jr et al. (2005)* |
| *B. nodoterga* | Central Serra do Mar | *ephippium* | Santana de Parnaíba | SP | ? | *Abegg et al. (2015)* |

Peer J

**Table 1** (*continued*)

| Species[a] | Relief units[b] | Group[c] | Locality[d] | State | Altitude | Source [e] |
|---|---|---|---|---|---|---|
| *B. toby* | Central Serra do Mar | *ephippium* | Ubatuba | SP | ? | *Silva, Campos & Sebben* (*2007*; as "*Brachycephalus* cf. *vertebralis*"), *Campos, Silva & Sebben* (*2010*; as "*Brachycephalus* cf. *vertebralis*"), *Campos* (*2011*) |
| *B. vertebralis* | Central Serra do Mar | *ephippium* | Cunha | SP | ? | *Pombal Jr* (*2001*) |
| *B. vertebralis* | Central Serra do Mar | *ephippium* | Parati | RJ | ? | *Silva, Campos & Sebben* (*2007*), *Campos, Silva & Sebben* (*2010*), *Campos* (*2011*) |
| *Brachycephalus* sp. 7 | Central Serra do Mar | *ephippium* | Biritiba-Mirim | SP | ? | *Silva, Campos & Sebben* (*2007*; as "*B. nodoterga*"), *Campos, Silva & Sebben* (*2010*; as "*Brachycephalus* sp. 1"), *Campos* (*2011*; as "BBM SP1") |
| *B. brunneus* | Southern Serra do Mar | *pernix* | Campina Grande do Sul | PR | ? | *Silva, Campos & Sebben* (*2007*), *Campos, Silva & Sebben* (*2010*), *Campos* (*2011*) |
| *Brachycephalus leopardus* | Southern Serra do Mar | *pernix* | Tijucas do Sul | PR | ? | *Campos* (*2011*; as "BTS PR") |
| *B. atelopoides* | Southern Serra da Mantiqueira | ? | Piquete | SP | ? | *Miranda-Ribeiro* (*1920*) |
| *Brachycephalus* sp. cf. *Brachycephalus* sp. 1 | Paranapiacaba | *didactylus* | Municipality of Ribeirão Grande | SP | ? | *Verdade et al.* (*2008*; as "*B. hermogenesi*") |
| *Brachycephalus* sp. cf. *Brachycephalus* sp. 1 | Paranapiacaba | *didactylus* | Municipality of Tapiraí | SP | ? | *Verdade et al.* (*2008*; as "*B. hermogenesi*"), *Condez, Sawaya & Dixo* (*2009*; as "*B. hermogenesi*") |
| *Brachycephalus* sp. cf. *Brachycephalus* sp. 1 | Paranapiacaba | *didactylus* | Municipality of Juquitiba | SP | ? | *Verdade et al.* (*2008*; as "*B. hermogenesi*") |
| *Brachycephalus* sp. cf. *Brachycephalus* sp. 1 | Central Serra do Mar | *didactylus* | Estação Ecológica Juréia-Itatins, municipality of Iguape | SP | ? | *Verdade et al.* (*2008*; as "*B. hermogenesi*") |

**Table 1** (*continued*)

| Species[a] | Relief units[b] | Group[c] | Locality[d] | State | Altitude | Source [e] |
|---|---|---|---|---|---|---|
| *Brachycephalus* sp. cf. *Brachycephalus* sp. 1 | Southern Serra do Mar | *didactylus* | Ilha do Cardoso, municipality of Cananéia | SP | ? | *Verdade et al.* (*2008*; as "possibly *B. hermogenesi*") |

**Notes.**

[a]Species and records are ordered by the following criteria: alphabetic order of groups and alphabetic order of species by groups.

[b]According to *IBAMA (2007)* with additional subdivisions (see 'Materials and Methods').

[c]For group definitions, see 'Materials and Methods'.

[d]Geographical coordinates are based on the datum WGS84.

[e]DZUP, Coleção de Herpetologia, Departamento de Zoologia, Universidade Federal do Paraná, state of Paraná, Brazil; MHNCI, Museu de História Natural Capão da Imbuia, Prefeitura Municipal de Curitiba, state of Paraná, Brazil.

[f]Excluded records represent vague localities, general reference of municipality of locality, and localities that were specified but could not be located or were questionable given exceedingly low associated altitude.

**Table 2** Number of species of *Brachycephalus* per relief unit.

| Relief unit[a] | Number of groups | Number of species (undescribed species included) |
|---|---|---|
| Pré-Litorâneas | 1 | 1 (0) |
| Serra da Mantiqueira | 2 | 3 (0) |
|     Northern Serra da Mantiqueira | 2 | 3 (0) |
|     Southern Serra da Mantiqueira | 1 | 1 (0) |
| Paraíba do Sul | 1 | 1 (0) |
| Serra do Mar | 3 | 30 (6) |
|     Northern Serra do Mar | 2 | 6 (1) |
|     Central Serra do Mar | 2 | 9 (1) |
|     Southern Serra do Mar | 2 | 16 (4) |
| Paranapiacaba | 1 | 1 (1) |
| Leste Catarinense | 1 | 3 (0) |

Notes.

[a] According to *IBAMA (2007)*, with additional subdivisions in the units Serra da Mantiqueira and Serra do Mar (see 'Materials and Methods').

A third undescribed species was assigned to the *didactylus* group because it represents a population to be split from *B. hermogenesi* (*Pie et al., 2013*), a representative species from the *didactylus* group. Finally, the other three populations were assigned to the *pernix* group because they were previously assigned to the "pernix clade" (*Pie et al., 2013*), whose members all belong to the *pernix* group (*Ribeiro et al., 2015*).

## RESULTS

The final dataset is composed of 333 records (from 73 distinc sources) of 120 localities (three of which representing novel unpublished records). These records involve the geographic occurrence of 28 formally known and six undescribed species, including 82 previous records that had been excluded based on the accuracy criteria indicated above (Table 1). Species were recorded in six relief units, the richest of them being Serra do Mar, with 30 species (six undescribed; Fig. 1), followed by Serra da Mantiqueira and Leste Catarinense, with three species each (Table 2). Northern Serra da Mantiqueira has records of all three species of this relief unit, while Southern Serra da Mantiqueira has records of only one species. In Serra do Mar, species richness increases from north to south, with records of six species (one undescribed) in Northern Serra do Mar, nine (one undescribed) in Central Serra do Mar, and 16 (four undescribed) in Southern Serra do Mar (Table 2; Fig. 1). Finally, Pré-Litorâneas, Paraíba do Sul, and Paranapiacaba relief units have records of only one species each (Paranapiacaba has records of an undescribed species; Table 2). The northernmost and southernmost records are 1,592 km distant from each other in a straight line; the species recorded in these limits are *B. pulex* and *B. fuscolineatus*, respectively (Table 1).

The *didactylus* group comprises four species (one undescribed) encompassing the broadest geographical distribution of all three groups, from the northernmost record for the genus (state of Bahia, with *B. pulex*) southward to the north of Santa Catarina state,

Relief units

Altitudinal quotas

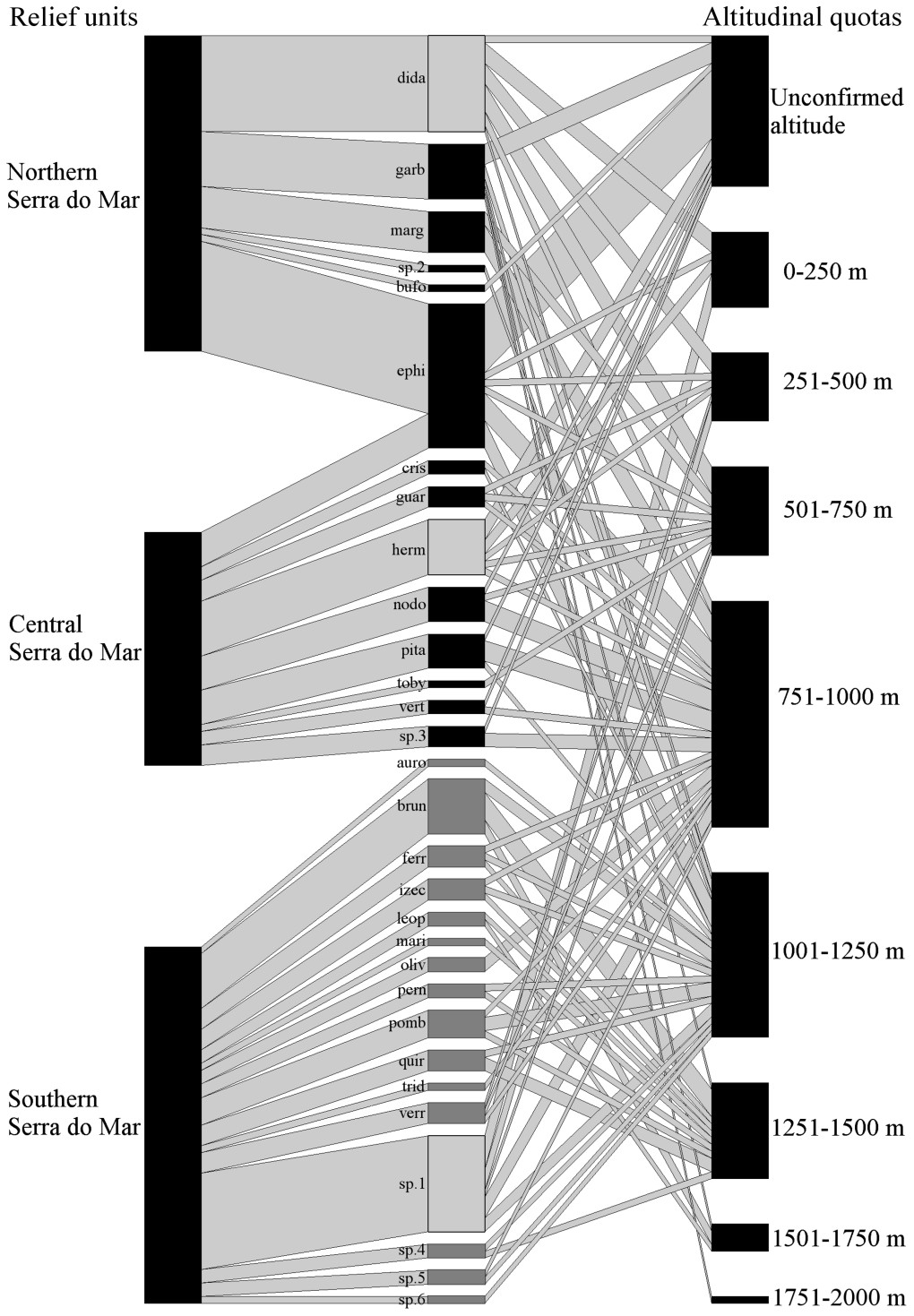

**Figure 1** Geographical and altitudinal distribution of *Brachycephalus* spp. according to their respective species group (light gray = *didactylus*, black = *ephippium*, and dark gray = *pernix* group) across the three sectors of the Serra do Mar relief unit. The species connections represent the occurrence in each sector of the relief unit and in altitudinal quotas. The width of the connections (in both directions) represents the proportional amount of citations in the bibliography for each sector and altitude quotas (wider = more citations). See Table 4 for the correspondence of the acronyms with the species name.

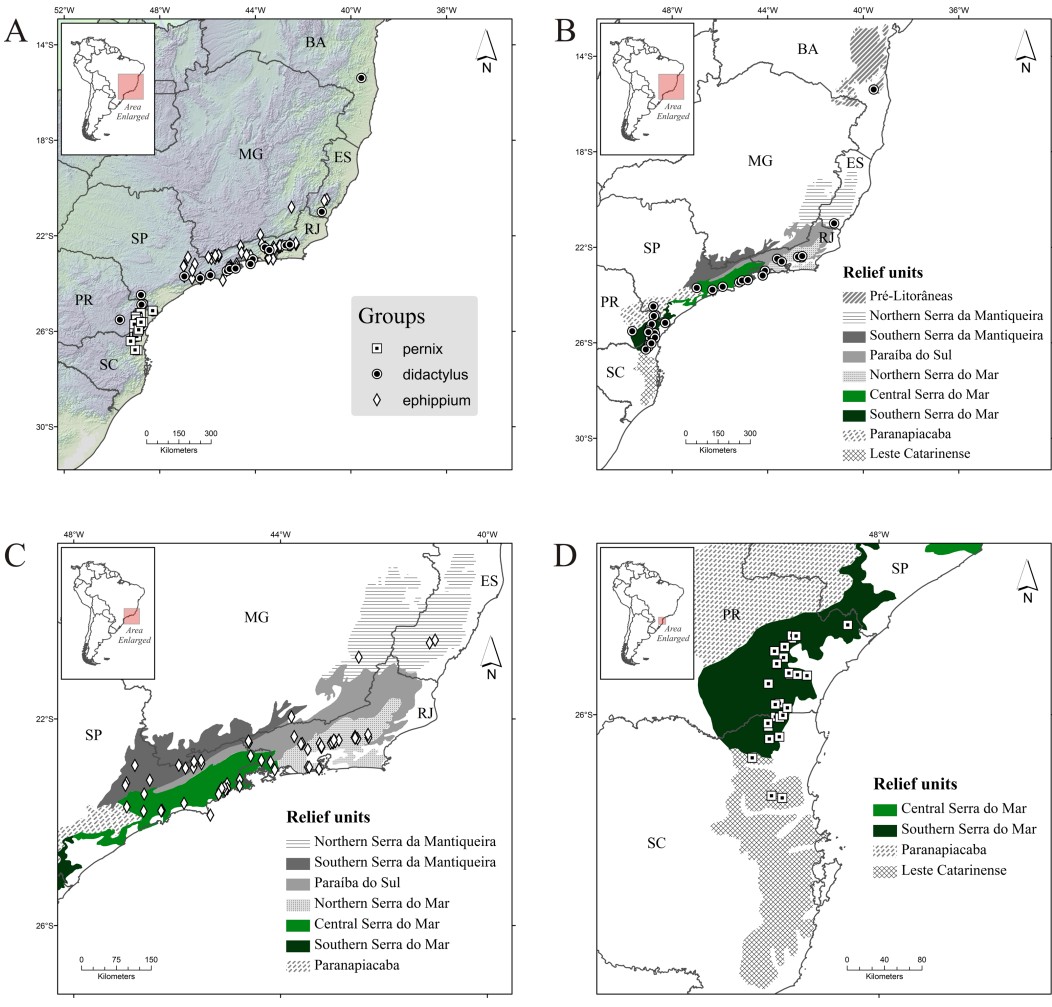

**Figure 2** Occurrence records of *Brachycephalus* spp. (A) all species; (B) species of the *didactylus* group; (C) species of the *ephippium* group; (D) species of the *pernix* group. Diamonds, records of the *ephippium* group; circles, records of the *didactylus* group; and squares, records of the *pernix* group. Abbreviations: BA, Bahia; MG, Minas Gerais; ES, Espírito Santo; RJ, Rio de Janeiro; SP, São Paulo; PR, Paraná; and SC, Santa Catarina. Relief units modified from *IBAMA (2007)*.

with *Brachycephalus* sp. 1 (Table 1; Fig. 2B). Three species occur in Serra do Mar (one undescribed), but only one species occurs in each of Northern, Central and Southern Serra do Mar (Southern Serra do Mar has records of one undescribed species; Table 3). Only one species occurs in the three remaining relief units with presence of this group (Pré-Litorâneas, Serra da Mantiqueira, and Paranapiacaba, with the latter being represented by an undescribed species; Table 3). There are no records of this group inland in the "Southern Serra da Mantiqueira" but there are records relatively further inland in "Paranapiacaba," in the state of Paraná (Fig. 2B).

The *ephippium* group comprises 13 species (two undescribed; Table 1). It has a more intermediate geographical distribution in relation to the overall latitudinal distribution of the genus, from the state of Espírito Santo to southern state of São Paulo (Fig. 2C);

**Table 3** Number of species of *Brachycephalus* per group in each relief unit.

| Relief unit[a] | Number of species per group (undescribed species included) | | |
|---|---|---|---|
| | *didactylus* | *ephippium* | *pernix* |
| Pré-Litorâneas | 1 (0) | 0 | 0 |
| Serra da Mantiqueira | 1 (0) | 2 (0) | 0 |
| Northern Serra da Mantiqueira | 1 (0) | 2 (0) | 0 |
| Southern Serra da Mantiqueira | 0 | 1 (0) | 0 |
| Paraíba do Sul | 0 | 1 (0) | 0 |
| Serra do Mar | 3 (1) | 12 (2) | 15 (3) |
| Northern Serra do Mar | 1 (0) | 5 (1) | 0 |
| Central Serra do Mar | 1 (0) | 8 (1) | 0 |
| Southern Serra do Mar | 1 (1) | 0 | 15 (3) |
| Paranapiacaba | 1 (1) | 0 | 0 |
| Leste Catarinense | 0 | 0 | 3 (0) |

**Notes.**

[a] According to *IBAMA (2007)*, with additional subdivisions in the units Serra da Mantiqueira and Serra do Mar (see Materials and methods).

the species recorded in those extreme geographical limits are *B. alipioi* and *B. ephippium*, respectively. The group is represented in all relief units encompassed in this region (Serra da Mantiqueira, Paraíba do Sul, and Serra do Mar; Table 1; Fig. 2C). It is highly represented by species in Serra do Mar (12 species, two of which undescribed; Fig. 1), whereas in Serra da Mantiqueira there is records of two species and in Paraíba do Sul of only one (Table 3). Northern Serra da Mantiqueira has two species, while Southern Serra da Mantiqueira has only one (Table 3). Northern Serra do Mar has five species (one undescribed) and Central Serra do Mar has eight species (also with one undescribed); Southern Serra do Mar does not harbor any species of this group (Table 3; Fig. 1).

The *pernix* group is the most species-rich group, with 17 species (three undescribed; Table 1). It has the smallest geographical distribution, restricted to the states of Paraná and Santa Catarina, southern Brazil (Fig. 1), comprising the southernmost distribution of the genus, with *B. fuscolineatus* being recorded 87 km south of the border of the state of Paraná; the northernmost species is *B. tridactylus* (Table 1). Per relief unit, 15 species (three undescribed) occurs in Southern Serra do Mar and three in Leste Catarinense (Table 3; Fig. 1).

We were able to determine the extent of occurrence of 17 species (51.5% of all species with records at precise locations; Table 4). Two species of the *didactylus* group have the highest estimated extent of occurrence, with over 500 thousand ha, namely *Brachycephalus* sp. 1 and *B. hermogenesi*. On the other hand, estimates of extents of occurrence for all remaining species did not reach 20 thousand ha. With proportionally intermediate extent of occurrence, there are tree species from the *ephippium* group (*B. garbeanus*, *B. margaritatus*, and *B. pitanga*), with estimates between 2–19 thousand ha, and five species from the *pernix* group (*B. olivaceus*, *B. brunneus*, *B. ferruginus*, *Brachycephalus* sp. 6, and *B. quiririensis*), with estimates between 1–6 thousand ha (Table 4). There are four species with small extent

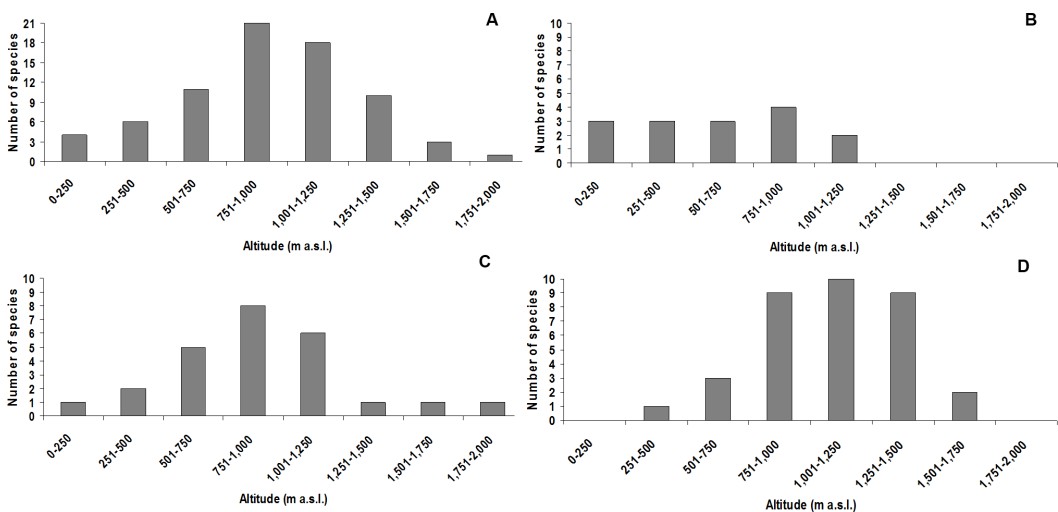

**Figure 3** Species richness of *Brachycephalus* in each altitudinal class (including undescribed species). (A) all species; (B) species of the *didactylus* group; (C) species of the *ephippium* group; (D) species of the *pernix* group.

of occurrence, between 100–500 hectares, one from the *didactylus* group—*B. pulex*—and three from the pernix group—*B. pernix*, *B. izecksohni*, and *B. leopardus* (Table 4). Finally, there are three species from the *pernix* group with highly reduced extents of occurrence, namely *B. tridactylus*, *B. fuscolineatus*, and *B. boticario*, with 41.42 ha, 23.63 ha, and 11.07 ha, respectively (Table 4).

*Brachycephalus* species are found from sea level to at least 1,900 m a.s.l. (Tables 1 and 3). The *ephippium* group was characterized by species with the broadest altitudinal amplitude (1,700 m; Table 4). The *didactylus* and *pernix* groups had similar altitudinal amplitude (1,110 m and 1,185 m, respectively; Table 4). However, the altitudinal occurrence of the *didactylus* group started at sea level, whereas the *pernix* group began at mid-elevations (455 m a.s.l.; Table 4).

The altitudinal class with the highest species richness was 751–1,000 m a.s.l., with 21 species (three undescribed; Fig. 3A). The highest species richness for the *didactylus* group was also found between 751–1,000 m a.s.l., including all four species of the group (one undescribed; Fig. 3B). The highest richness of the *ephippium* group was evenly recorded between 751–1,000 m, comprising eight species (one undescribed; Fig. 3C). Finally, most species of the *pernix* group were found between 1,001–1,250 m a.s.l., totalizing 10 species (three undescribed; Fig. 3D).

The species with the broadest altitudinal range was *B. didactylus* (1,075 m; starting at 35 m a.s.l.) and *Brachycephalus* sp. 1 (1,035 m; starting at 25 m a.s.l.), both included in the *didactylus* group, and *B. ephippium* (1,050 m; starting from 200 m a.s.l.), included in the *ephippium* group (Table 4). The largest altitudinal occurrence amplitude for species of the *pernix* group was recorded in *B. brunneus* (535 m; starting at 1,095 m a.s.l.; Table 4). The lowest records for the *pernix* group were at 845 m a.s.l. in the state of Paraná and at 455 m a.s.l. in the state of Santa Catarina, southern Brazil (Table 1). This illustrates the tendency

**Table 4** **Altitudinal distribution and "extent of occurrence"** (*sensu IUCN, 2012*) **of** *Brachycephalus* **spp.** Only confirmed altitudinal records are considered. Extent of occurrence by altitude (Alt) or by an adaptation of the minimum convex polygon (MCP; see text for details).

| Species[a] | Altitudinal distribution (m a.s.l.) | | Extent of occurrence (ha) | |
|---|---|---|---|---|
| | Range | Amplitude | Alt | MCP |
| *didactylus* group | 0–1,110 | 1,110 | – | – |
| *B. didactylus* (dida) | 35–1,110 | 1,075 | ? | ? |
| *B. hermogenesi* (herm) | 0–900 | 900 | – | 567,589.87 |
| *B. pulex* | 800–930 | 130 | 488.25 | – |
| *Brachycephalus* sp. 1 (sp.1) | 25–1,060 | 1,035 | – | 778,458.42 |
| *ephippium* group | 200–1,900 | 1,700 | – | – |
| *B. alipioi* | 1,070–1,100 | 30 | ? | ? |
| *B. crispus* (cris) | 800–1,100 | 300 | ? | ? |
| *B. ephippium* (ephi) | 200–1,250 | 1,050 | ? | ? |
| *B. garbeanus* (garb) | 1,130–1,900 | 770 | – | 12,268.00 |
| *B. guarani* (guar) | 500–900 | 400 | ? | ? |
| *B. margaritatus* (marg) | 600–980 | 380 | – | 18,272.87 |
| *B. nodoterga* (nodo) | 700–945 | 245 | ? | ? |
| *B. pitanga* (pita) | 900–1,140 | 240 | – | 2,377.07 |
| *B. toby* (toby) | 750 | 0 | ? | ? |
| *B. vertebralis* (vert) | 900 | 0 | ? | ? |
| *Brachycephalus* sp. 2 (sp.2) | 1,100–1,200 | 100 | ? | ? |
| *Brachycephalus* sp. 3 (sp.3) | 800–825 | 25 | ? | ? |
| *pernix* group | 455–1,640 | 1,185 | – | – |
| *B. auroguttatus* (auro) | 1,070–1,100 | 30 | ? | ? |
| *B. boticario* | 755–795 | 40 | 11.07 | – |
| *B. brunneus* (brun) | 1,095–1,630 | 535 | 5,687.08 | – |
| *B. ferruginus* (ferr) | 965–1,470 | 505 | 5,475.51 | – |
| *B. fuscolineatus* | 640–790 | 150 | 23.63 | – |
| *B. izecksohni* (izec) | 980–1,340 | 360 | 350.43 | – |
| *B. leopardus* (leop) | 1,400–1,640 | 240 | 176.73 | – |
| *B. mariaeterezae* (mari) | 1,265–1,270 | 5 | ? | ? |
| *B. olivaceus* (oliv) | 690–985 | 295 | – | 12,531.64 |
| *B. pernix* (pern) | 1,135–1,405 | 270 | 432.10 | – |
| *B. pombali* (pomb) | 845–1,300 | 455 | ? | ? |
| *B. quiririensis* (quir) | 1,240–1,380 | 140 | 1,338.97 | – |
| *B. tridactylus* (trid) | 805–910 | 105 | 41.42 | – |
| *B. verrucosus* (verr) | 455–945 | 490 | ? | ? |
| *Brachycephalus* sp. 4 (sp.4) | 1,250–1,320 | 70 | ? | ? |
| *Brachycephalus* sp. 5 (sp.5) | 1,000–1,080 | 80 | ? | ? |
| *Brachycephalus* sp. 6 (sp.6) | 1,095–1,160 | 65 | 2,211.54 | – |

**Notes.**
[a] Abreviation in parentheses refers to species acronyms in Fig. 1.

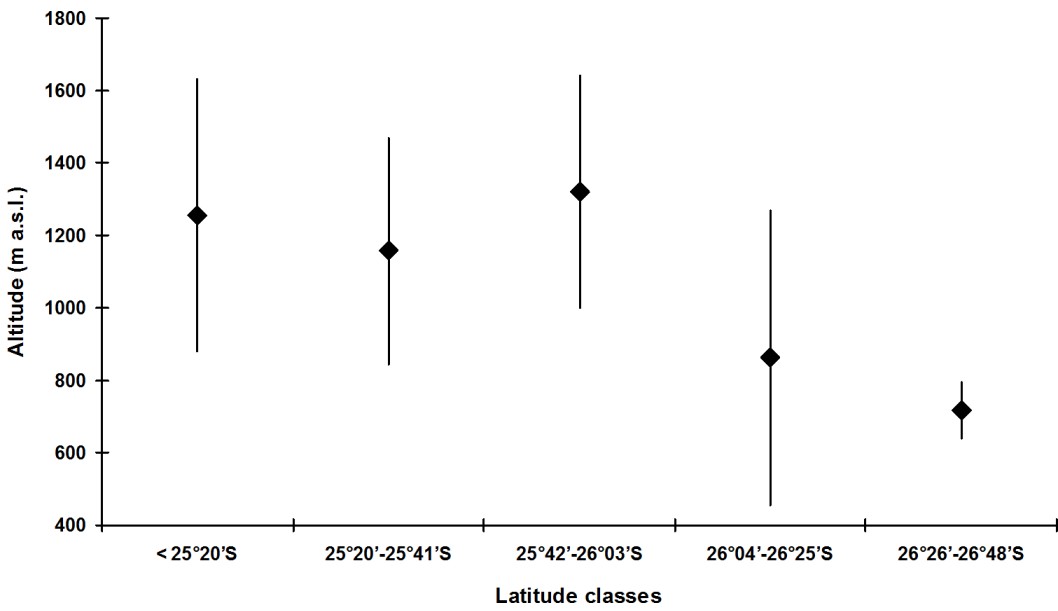

**Figure 4** **Altitudinal amplitude records of *Brachycephalus* spp. from the *pernix* group binned into latitudinal classes (including undescribed species).** Diamonds indicate median values.

for the altitudinal distribution of species of the *pernix* group to decrease with increasing latitude (Fig. 4).

Syntopy in *Brachycephalus* is rare, being reported for the *didactylus* and *ephippium* groups in state of Rio de Janeiro and São Paulo, and for the *didactylus* and *pernix* groups in the state of Paraná and Santa Catarina (Table 1). Between *didactylus* and *ephippium* groups, syntopy has been reported in four localities: between *B. didactylus* and *B. ephippium* in the Reserva Ecológica Rio das Pedras, Rio de Janeiro (*Carvalho-e-Silva, Silva & Carvalho-e-Silva, 2008*; *Almeida-Santos et al., 2011*; *Rocha et al., 2013*); between *B. didactylus* and *B. margaritatus* in the Sacra Família do Tinguá, Rio de Janeiro (*Izecksohn, 1971*; *Pombal Jr, 2001*; *Ribeiro et al., 2005*; *Alves et al., 2006*; *Alves et al., 2009*; *Silva, Campos & Sebben, 2007*; *Verdade et al., 2008*; *Clemente-Carvalho et al., 2009*; *Campos, 2011*; *Pombal Jr & Izecksohn, 2011*; *Pie et al., 2013*); between *B. didactylus* and *Brachycephalus* sp. 2 in Theodoro de Oliveira, Rio de Janeiro (*Siqueira et al., 2011*; *Siqueira, Vrcibradic & Rocha, 2013*); and between *B. hermogenesi* and *B. ephippium* in the Reserva Florestal de Morro Grande, São Paulo (*Dixo & Verdade, 2006*; *Verdade et al., 2008*). Between *didactylus* and *pernix* groups, two cases of syntopy have been recorded (M Bornschein, pers. obs., 2016), one between *Brachycephalus* sp. 1 (*didactylus* group) and *B. tridactylus*, in Reserva Particular do Patrimônio Natural Salto Morato, at 900 m a.s.l., and the another one between *Brachycephalus* sp. 1 and *B. olivaceus* at Castelo dos Bugres, between 800–835 m a.s.l. (M Bornschein, pers. obs., 2011; Table 1).

Sympatry and possibly syntopy also between *didactylus* and *ephippium* groups was reported in the state of São Paulo in two localities (Table 1): between *B. hermogenesi* and *Brachycephalus* sp. 3 in the Reserva Biológica do Alto da Serra de Paranapiacaba (*Verdade et al., 2008*; *Pie et al., 2013*) and between *B. hermogenesi* and *B. nodoterga* in the Estação

Biológica de Boracéia (*Pombal Jr, Wistuba & Bornschein, 1998*; *Pombal Jr, 2001*; *Pombal Jr, 2010*; *Ribeiro et al., 2005*; *Alves et al., 2006*; *Alves et al., 2009*; *Pimenta, Bérnils & Pombal Jr, 2007*; *Verdade et al., 2008*; *Haddad et al., 2010*; *Pombal Jr & Izecksohn, 2011*; *Pie et al., 2013*). A sympatry may occur between species of a single group, particularly the *ephippium* group, in the state of Rio de Janeiro (Table 1): between *B. bufonoides* and *B. garbeanus* in the Serra de Macaé (*Miranda-Ribeiro, 1920*; *Pombal Jr, 2010*; *Pombal Jr & Izecksohn, 2011*) and between *B. ephippium* and *B. vertebralis* in the Pedra Branca, municipality of Parati (*Pombal Jr, 2001*; *Pombal Jr, 2010*; *Clemente-Carvalho et al., 2009*; *Pombal Jr & Izecksohn, 2011*; *Pie et al., 2013*). Both need confirmation (see *Pombal Jr, 2001*).

## DISCUSSION

The northernmost and southernmost relief units are poorly represented by records of *Brachycephalus*, namely Pré-Litorâneas and Leste Catarinense, respectively. This suggests the possibility that the genus occurs farther north and south from the most extreme records known today (see Fig. 2). Interestingly, the potential distribution of *Brachycephalus* through environmental niche modeling (*Pie et al., 2013*) is consistent with the possibility of a wider distribution of the genus to the south of the current distribution, spreading widely through the Leste Catarinense relief unit (see Figs. 3A, 3B, 3E, 3F and 4A of *Pie et al. (2013)*). Forests at lower altitudes in the Pré-Litorâneas region are different from those in mountains at higher altitudes, displaying, for example, little moss on tree trunks and scarce epiphytic plants (MRB & RB-L, per. obs.). We believe it is possible that the occurrence of the genus in this relief unit is restricted to mountains. Examples of locations of scarce humid mountains of this region can be seen in *Maurício et al. (2014)*, who mapped the occurrence of an endemic mountain bird to the Pré-Litorâneas relief unit, the *Scytalopus gonzagai*. The environmental niche modeling of *Brachycephalus* indeed indicates a small area of suitable habitats in this region (see Fig. 3B of *Pie et al., 2013*).

The occurrence of *B. ephippium* in distinct relief units should be the subject of further scrutiny. In particular, populations under that name might indeed represent cryptic species that would have to be split (see also *Dixo & Verdade, 2006*; *Silva, Campos & Sebben, 2007*; *Clemente-Carvalho et al., 2008*; *Clemente-Carvalho et al., 2011a*; *Campos, Silva & Sebben, 2010*; *Campos, 2011*; *Siqueira, Vrcibradic & Rocha, 2013*; *Trevine et al., 2014*). Similar conditions were detected in a montane bird, *Scytalopus speluncae* (taxonomy according to *Maurício et al. (2010)*), which has to be split into several new species. In fact, the distribution of *S. speluncae* is congruent with that of *B. ephippium* in some relief units (*Mata et al., 2009*). The occurrence of *B. ephippium* around the city of São Paulo is of particular interest, given this region is characterized as a contact between Serra da Mantiqueira and Serra do Mar relief units (*IBGE, 1993*).

For a long time, all known species of *Brachycephalus* of the *pernix* group had their geographical distribution limited to a single "Serra," i.e., in a massif range with high altitudes, isolated from other "Serras" by areas of lower altitudes. *Pie et al. (2013)* were the first to report the occurrence of more than one species of the *pernix* group in a single "Serra," and here we report on other similar situations. In addition, the present study is

the first to report the occurrence of a species of this group distributed in two "Serras." This is the case of *B. brunneus*, distributed in Serra dos Órgãos (not to be confused with the homonym "Serra" of the state of Rio de Janeiro) and Serra da Graciosa (Table 1). Distributions of more than one species in a single "Serra" involve the cases of *B. ferruginus* and *Brachycephalus* sp. 4, distributed in Serra do Marumbi, and *B. leopardus*, *B. auroguttatus*, and *B. quiririensis*, distributed in Serra do Araçatuba/Serra do Quiriri, a single mountain massif in the border of the states of Paraná and Santa Catarina (Table 1). These cases do not represent syntopy because all of them are isolated from each other by more than 3 km. The ocurrences of two species of the *ephippium* group in a single Serra are known for long time, i.e., 1920 (involving *B. bufonoides* and *B. garbeanus*; *Miranda-Ribeiro (1920)*) and 2001 (involving *B. ephippium* and *B. vertebralis* *Pombal Jr (2001)*).

It is interesting to note that the species with low altitudinal records (and also with high altitudinal amplitude of occurrence) are the ones of greatest extent of occurrence measured, perhaps indicating that its relatively plastic condition provide them a major opportunity to establish through a continuous Atlantic Forest. In the other way, species with medium to high lower limit of altitudinal occurrence, and also with moderate altitudinal amplitude of occurrence, have very reduced extent of occurrence, indicating proportionaly reduced plasticity and subjection of environmental and topographic constrains. In a study on a Neotropical bird, *Reinert, Bornschein & Firkowski (2007)* showed that a very detailed estimate of the species extent of occurrence resulted in an area seven times greater than the area of occupancy, which underscores how much the extent of occurrence of *Brachycephalus* could be overstimated. In comparison, the small extent of occurrence of the *Brachycephalus* is not the smaller "range" of species in the world; among the smaller, there are an isopod and a fish, with less than 0.01 ha of occurrence, each (*Brown, Stevens & Kaufman, 1996*). However, they are between the smallest "ranges" of vascular plants and fishes of the world (<100 ha) and much smaller than the smallest "ranges" of birds and mammals of the world (∼1,000,000 ha; *Brown, Stevens & Kaufman (1996)*).

The altitudinal records indicate that the *pernix* and *ephippium* groups are indeed "montane groups" (Figs. 1 and 3), even though the altitudinal amplitude of species of the *ephippium* group might be underestimated. Occasional records below 500 m a.s.l. for species of both montane groups do not mean that all "montane species" of the genus normally occur at lower altitudes, with the absence of lower records resulting from sampling bias. Rather, one could expect that species of the *pernix* group at higher latitudes would tend to be recorded at lower altitudes due to climatic compensation. Moreover, local records at lower altitudes of species of montane groups can be due to particular local conditions of high moisture resulting from peculiar topographical features. Regardless of the extreme altitudinal records, each species must have an altitudinal range that corresponds to its climatic optimum (see Fig. 1). Studies estimating abundance along altitudinal transects can be used not only to determine altitudinal ranges but also the climatic optimum for each species. These data are of particular interest in the context of long-term studies to assess the impact of climatic changes, with gradual modification in the abundances and even changes in the altitudinal range being expected over time (see *Corn, 2005*; *Jump, Huang & Chou, 2012*).

All records of *pernix* group species in the state of Paraná were either made in cloud forests ("Floresta Ombrófila Densa Alto-montana" *sensu Veloso, Rangel-Filho & Lima (1991)*) or in the transition zones between cloud forest and montane forests ("Floresta Ombrófila Densa Montana"). The only exception is *B. tridactylus*, which occurs in montane forests (this study, since there is no available description for the forest type in the species description by *Garey et al., 2012*). Cloud or elfin forests are pygmy tropical forests that occur in highly humid regions at the cloud level, between 1,000 to 2,500 m a.s.l. (*Walter, 1977*). In the state of Paraná, these forests occur from 900 m to about 1,850 m a.s.l. (*Struminski, 1997*; *Bornschein et al., 2012*). The lowest record of *B. pombali* (845 m a.s.l.), in the transition between cloud forest and montane forest, probably represents the lowest altitude of this transition in the state of Paraná. In Santa Catarina, species from the *pernix* group occur in cloud forests, montane forests, and in the transition between them. The only known exception of forest habitat in the *pernix* group involves the observation of *B. izecksohni*, also in herbaceous montane fields ("campos de altitude" or "refúgio vegetacional", *sensu Veloso, Rangel-Filho & Lima (1991)*; M Bornschein, pers. obs., 2000).

The few cases of syntopy between species, not only show little altitudinal overlap, but also involve species of different groups, consistent with the notion that closely related species remain allopatric. Localities with sympatry and syntopy are particularly interesting for further studies related to the altitudinal distribution of species. On the other hand, sympatry between species of the groups *ephippium* and *pernix* is not expected, given that a gap of 208 km occurs between species of these groups. It is interesting to note that only the montane species groups, i.e., *ephippium* and *pernix*, are highly species-rich, yet they tend not to show sympatry (although there are some species distributed very close to each other, and that there are cases of distinct species occurring in a single "Serra"). We believe that these conditions suggest that these groups evolved in allopatry and/or parapatry while the Atlantic Forest experienced retraction and isolation as islands between grasslands. During the Quaternary, climatic conditions changed cyclically from cold and dry to warmer and wet periods, and these shifts were responsible for the cyclical retraction and expansion of forest coverage, respectively (e.g., *Behling & Lichte, 1997*; *Behling et al., 2002*; *Behling, Pillar & Depatta, 2005*; *Behling et al., 2007*; *Behling, 2007*; *Langone et al., 2008*; *Behling & Safford, 2010*; *Enters et al., 2010*; *Hessler et al., 2010*). In colder climates, montane forests may have dispersed to lower altitudes, allowing species of *Brachycephalus* to disperse. In subsequent warmer and wet climate, the montane forest may have dispersed to higher altitudes, isolating populations of *Brachycephalus* and favoring speciation.

In relation to the species distributed close to each other or in single "Serras", without apparent geographic barriers, it is possible that the expansion and retraction of forests did not occur homogeneously across altitudinal quotas, but it might have occurred as isolated patches even in restricted areas. In areas where climatic conditions favored the return of a vast forest cover, geographically close species would have lost the corresponding environmental barriers, remaining allopatric probably due to ecological requirements. The most notable case of proximity between species occurs in northeastern state of São Paulo, where higher mountains are absent and five species occur in a single plateau (*B. crispus*,

*B. guarani*, *B. pitanga*, *B. toby*, and *B. vertebralis*). In this region, the forests may have been confined in valleys during the periods of retraction in the forest distribution.

In conclusion, *Brachycephalus* is restricted to the eastern portion of the Atlantic Forest biome, with species occurring in allopatry or at least in parapatry, with rare cases of syntopy. The species of the genus are segregated in three groups of species, with one of them (*didactylus*) including species that respond differently to altitude, occurring in lower altitudes and having greater geographic distributions, being much more ecologicaly tolerant and plastic. Species from the remaining groups (*ephippium* and *pernix*) depend on high altitude conditions and can be found locally in lower altitudes, probably in response to particular microclimatic conditions. Sympatric species include members of distinct species groups and are in contact only in the altitudinal limits of each other (higher limit for the "lowland" species and lower limit for the "montane" species). Apart from a few species, most of them have restricted extents of occurrence, to the point that many are microendemic, occurring in mountain tops with a total extent of occurrence comparable to the smallest range from species around world. The microhabitat requirements of the species can prevent more realistic estimatives of extent of occurrence, with traditional methodologies possibly resulting in substantial bias. This same feature makes the species of particular interest to monitor climate change through their abundances along altitudinal gradients. The genus also can be used as a model to infer historical environmental changes due to climatic changes, given that speciation in *Brachycephalus* species is thought to have resulted from vicariant processes following upward dispersal of montane forests through altitudinal ranges, in past warm and wet climates.

## ACKNOWLEDGEMENTS

We thank Magno V. Segalla and Camila R. Alves for valuable assistance during fieldwork Claudia Golec helped in the compilation of bibliography. Two anonymous referees improved the manuscript with valuable comments and suggestions.

### Funding

MRB received a PRODOC grant from CAPES (project 2599/2010). RBL is supported by fellowships from CNPq/MCT (141823/2011–9). MRP is supported by grant from CNPq/MCT (571334/2008–3). Fieldwork during 2011 and 2012 was partially funded by Fundação Grupo O Boticário de Proteção à Natureza (trough the project 0895_20111 conducted by Mater Natura—Instituto de Estudos Ambientais). The funders had no role in study design, data collection and analysis, decision to publish, or preparation of the manuscript.

### Grant Disclosures

The following grant information was disclosed by the authors:
CAPES: 2599/2010.

CNPq/MCT: 141823/2011–9, 571334/2008–3.
Fundação Grupo O Boticário de Proteção à Natureza: 0895_20111.

## Competing Interests

Marcio Pie is an Academic Editor for PeerJ. Sérgio A.A. Morato is employed by STCP Engenharia de Projetos Ltda.

## Author Contributions

- Marcos R. Bornschein conceived and designed the experiments, performed the experiments, analyzed the data, contributed reagents/materials/analysis tools, wrote the paper, prepared figures and/or tables, reviewed drafts of the paper, compiled the data.
- Carina R. Firkowski, Leandro Corrêa, Luiz F. Ribeiro and Sérgio A.A. Morato analyzed the data, reviewed drafts of the paper.
- Ricardo Belmonte-Lopes and Andreas L.S. Meyer analyzed the data, prepared figures and/or tables, reviewed drafts of the paper.
- Reuber L. Antoniazzi-Jr. analyzed the data, prepared figures and/or tables.
- Bianca L. Reinert and Felipe A. Cini analyzed the data.
- Marcio R. Pie analyzed the data, contributed reagents/materials/analysis tools, reviewed drafts of the paper.

## Field Study Permissions

The following information was supplied relating to field study approvals (i.e., approving body and any reference numbers):

Collection and research permits for this study were issued by ICMBIO (22470–1 and 22470–2), Instituto Ambiental do Paraná (permit 355/11), and Fundação Municipal do Meio Ambiente of Joinville (permit 001/11), for the regional permits.

## Data Availability

The raw data in this article is included in the tables and figures.

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
