# Peer review of "Geographical and altitudinal distribution of Brachycephalus (Anura: Brachycephalidae) endemic to the Brazilian Atlantic Rainforest"

_PeerJ, doi:10.7717/peerj.2490_

## Round 0.1 · original submission · Major Revisions

Two referees have found that your manuscript is a valuable and interesting contribution but they have also found a series of questions that you should carefully address before resubmiting. Please make modifications and answer the referees comments.

Reviewer 1 ·

Basic reporting

This is a well written manuscript analyzing data on the altitudinal range of one of the most exciting groups of frogs, the miniaturized genus Brachycephalus. The manuscript is quite wordy and long, but I feel this is justified since it gives a nice review of what is know on the distribution, systematics, and natural history of these frogs.

I have two main issues with this manuscript:

The first is that the manuscript needs to state more clearly which are the novel aspects of this study compared to previously published work, in particular Pie et al. in Ecological Modelling. This does not involve just adding one or two numbers but should be added at numerous parts of the manuscrit including the Abstract, Introduction, Materials and Methods, and Discussion. It extends mentioning clearly how many new locality records are being added here, providing a justification for including the lengthy Table 1 (was such a detailed list of localities already given in Pie et al.?), summarizing the work and results of Pie et al. more clearly (what is said in that aper about range sizes?).

My second point is that the manuscript would strongly benefit from estimating maximum range sizes of the different species of Brachycephalus. It is an important point that tiny amphibians such as Brachycephalus have very small range sizes, but exceptions do occur (see paper by Gehara et al. 2014 in PLOS One, showing that some species of the small-sized Denropsophus minutus complex have vast ranges). In order to facilitate comparisons, it would be important to obtain range size estimates for the various Brachycephalus species. Minimum range sizes for species with single locality records are difficult to estimate of course, but maximum range sizes could be estimated either from (1) clipped ENMs, or (2) simply from minimum spanning polygons. I would like to see in this paper, estimates of the range sizes of those species with more than 2 distribution points, in orer to provide information of maximum ranges in this miniaturized genus.

A further point is that in the paragraph spanning from lines 98-104, the two listed references are insufficient. There are numerous influential works on refugia in the Atlantic forests, and at least 5-6 of these should be cited here. For amphibians this include works of Ana Carnaval and others, but there are many other important ones for other animal groups.

In addition, I encountered many small small issues of awkward phrasing that require corection or revision. I am in the following listing the ones that I stumbled upon, after the respective line number. But please proofread the whole manuscript one more time for errors of the same kind as I am convinced I have overlooked some.

Line 52, of the pernix group
54, between species is
66, dating back to the Paleocene
89, at the border
90 In the state
93, in the state
142 the didactylus and ephippium groups were not monophyletic.
144, a possible reason why
167, because the holotype is missing [this whole sentence should better be rephrased]
171, I don't understand this sentence: "convergent in respect to species composition" - please rephrase

Experimental design

I have no comments on the experimental design, and the PeerJ conditions in this section are fulfilled.

Validity of the findings

I have no comments on the validity of the findings and the PeerJ conditions in this section are fulfilled.

Additional comments

All general comments are included in my basic reporting.

Reviewer 2 ·

Basic reporting

No comments

Experimental design

No comments

Validity of the findings

No comments

Additional comments

This article contains a lot of very useful, publishable data on the distribution of species of Brachycephalus. The English is quite good and I have only one minor comment.

The paper begins by talking about mountains and the frogs are not introduced until line 102 ("Brachycephalus is a genus..."). Because the paper is about the frogs, they should be introduced first. Also, the intro contains a lot of discussion and citations of ecological data and authors, taxonomic names and authors, and past relationship studies. Because this is a distributional paper, much of that can be omitted and just cite the article with the tree being used for relationships. Authors of species names are not needed because this is not a taxonomic paper per se.

Line 395: "sympathy" should be "sympatry" I think.

---

## Round 0.2 · accepted · Accept

Thanks to your modifications, I am happy to recommend your manuscript for publication

Reviewer 1 ·

Basic reporting

The authors have done an excellent job revising the manuscript. All of my previous suggestions were adequately incorporated. I look forward to seeing this paper published.

I have one note to the authors that maybe they can check for one of their future papers on Brachycephalus. This concerns the name "Brachycephalus atelopoide". I think this should be changed to B. atelopoides. The name was originally coined as subspecies/variety of B. ephippium. In this species name, "ephippium" is neuter and probably needs to be interpreted as invariable noun in apposition. Then, "atelopoide" was formed neuter because it referred to the original species name. If now considered a distinct species, the species name which to me is clearly an adjective, should probably be changed to agree in gender with the genus name. I know Frost lists the species as atelopoide, but he is not an expert in Latin/Ancient Greek. I might be wrong also, and "atelopoide" would be no adjective but a noun in apposition. Probably this should be consulted with a specialist in nomenclature, maybe Alain Dubois, and corrected if adequate.

Experimental design

No comments.

Validity of the findings

No comments

Additional comments

No comments